# *Actinidia deliciosa* Extract as a Promising Supplemental Agent for Hepatic and Renal Complication-Associated Type 2 Diabetes (In Vivo and In Silico-Based Studies)

**DOI:** 10.3390/ijms241813759

**Published:** 2023-09-06

**Authors:** Eman Fawzy El Azab, Saleha Y. M. Alakilli, Abdulrahman M. Saleh, Hassan H. Alhassan, Hamad H. Alanazi, Heba Bassiony Ghanem, Sara Osman Yousif, Heba Abu Alrub, Nahla Anber, Elyasa Mustafa Elfaki, Alneil Hamza, Shaymaa Abdulmalek

**Affiliations:** 1Department of Clinical Laboratory Sciences, College of Applied Medical Sciences at Al-Qurayyat, Jouf University, Al-Qurayyat 77454, Saudi Arabia; hhalanzi@ju.edu.sa (H.H.A.); soyousif@ju.edu.sa (S.O.Y.); htaburub@ju.edu.sa (H.A.A.); eelfaki@ju.edu.sa (E.M.E.); aahamza@ju.edu.sa (A.H.); 2Department of Biological Sciences, Faculty of Sciences, King Abdulaziz University, Jeddah 23761, Saudi Arabia; salakilli@kau.edu.sa; 3Pharmaceutical Medicinal Chemistry & Drug Design Department, Faculty of Pharmacy (Boys), Al-Azhar University, Cairo 11884, Egypt; abdo.saleh240@azhar.edu.eg; 4Department of Clinical Laboratory Sciences, College of Applied Medical Sciences, Jouf University, Sakaka 72341, Saudi Arabia; h.alhasan@ju.edu.sa (H.H.A.); hbghanem@ju.edu.sa (H.B.G.); 5Medical Biochemistry Department, Faculty of Medicine, Tanta University, Tanta 31527, Egypt; 6Department of Clinical Chemistry, Faculty of medical Laboratory Sciences, Sudan University of Science and Technology, Khartoum 13311, Sudan; 7Emergency Hospital, Mansoura University, Mansoura 35516, Egypt; nahlaamedanber@mans.edu.eg; 8Biochemistry Department, Faculty of Science, Alexandria University, Alexandria 21511, Egypt; shaymaa.abdulmalek@alexu.edu.eg

**Keywords:** *Actinidia deliciosa*, adipocytokines, insulin resistance, mTOR, SIRT-1 pathway, PTEN, caffeic acid, melezitose

## Abstract

Type 2 diabetes (T2D) is a chronic metabolic condition associated with obesity, oxidative stress-mediated inflammation, apoptosis, and impaired insulin signaling. The utilization of phytochemical therapy generated from plants has emerged as a promising approach for the treatment of diabetes and its complications. Kiwifruit is recognized for its substantial content of antioxidative phenolics. Therefore, this work aimed to examine the effect of *Actinidia deliciosa* (kiwi fruit) on hepatorenal damage in a high-fat diet (HFD) and streptozotocin (STZ)-induced T2D in rats using in vivo and in silico analyses. An increase in hepatic and renal lipid peroxidation was observed in diabetic rats accompanied by a decrease in antioxidant status. Furthermore, it is important to highlight that there were observable inflammatory and apoptotic responses in the hepatic and renal organs of rats with diabetes, along with a dysregulation of the phosphorylation levels of mammalian target of rapamycin (mTOR), protein kinase B (Akt), and phosphoinositide 3-kinase (PI3K) signaling proteins. However, the administration of kiwi extract to diabetic rats alleviated hepatorenal dysfunction, inflammatory processes, oxidative injury, and apoptotic events with activation of the insulin signaling pathway. Furthermore, molecular docking and dynamic simulation studies revealed quercetin, chlorogenic acid, and melezitose as components of kiwi extract that docked well with potential as effective natural products for activating the silent information regulator 1(SIRT-1) pathway. Furthermore, phenolic acids in kiwi extract, especially syringic acid, P-coumaric acid, caffeic acid, and ferulic acid, have the ability to inhibit the phosphatase and tensin homolog (PTEN) active site. In conclusion, it can be argued that kiwi extract may present a potentially beneficial adjunctive therapy approach for the treatment of diabetic hepatorenal complications.

## 1. Introduction

Diabetes mellitus (DM) is regarded as a major health concern in the modern world due to its broadened spectrum of physiological consequences and prevalence [1]. According to the International Diabetes Federation (IDF), 382 million people were diagnosed with diabetes globally in 2013, and 463 million were diagnosed in 2019; this is expected to rise to 592 million by 2025. Prolonged hyperglycemia connected with several chronic metabolic problems is referred to as DM complications. Elevated blood glucose levels in type 2 diabetes (T2D) patients can cause serious, even catastrophic, vascular consequences such as amputations, atherosclerosis, retinopathy, neuropathy, and nephropathy [2,3].

Based on previous research, T2D is a chronic metabolic condition where the most frequent source of reactive oxygen species (ROS) generation, the mitochondria, plays a crucial role [3]. The progress of diabetes leads to oxidative stress attributable to the formation of free radicals from protein glycosylation, glucose autooxidation, and polyol pathways. Persistent cellular harm results from decreased antioxidant levels and/or irregular increases in ROS levels [4]. According to research, oxidative stress [4], inflammatory cytokines, chemokines, apoptosis, autophagy, and ferroptosis all trigger noteworthy factors in the beginning and development of diabetic complications [5].

Anti-diabetic pharmaceuticals are available in a wide range of formulations, some of which function similarly, for example, by improving insulin resistance, controlling blood sugar levels, and reducing oxidative stress and inflammation [6]. The bulk of these diabetes drugs are ineffective and have several bad side effects, such as weight gain, drug resistance, dropsy, and high rates of secondary failure [3]. Therefore, more research is required to create low-toxicity, cost-effective anti-diabetic medicines and to diminish T2D consequences, particularly in long-term treatment. Plant-based drugs have recently acquired more attention due to their low toxicity, accessibility, and convenience of use, intending to avoid oxidative stress and inflammation while activating antioxidant defense systems to lessen diabetes-related cell damage and consequences [5,7].

Kiwifruit is a delicious berry that develops on woody vines called Actinidia, which has several different kinds. The *Actinidia deliciosa* species of green kiwifruit is one of those that are generally available. With high levels of the vitamins K, E, and C, as well as folate, carotenoids, potassium, fibers, and phytochemicals, including a range of polyphenols and flavonoids, kiwis are among the most nutrient-dense fruits [8]. The antihypertensive, hypocholesterolemic, and fibrinolytic effects of kiwifruit were validated by Hunter et al. [9] in vitro, underscoring the significance of bioactive components in cardiovascular protection. Additionally, studies have indicated that kiwifruit can help prevent coronary artery disease by reducing blood triglyceride levels [10]. Furthermore, kiwifruits have proven anti-proliferative capabilities against human lung, colon, stomach [11], and pancreatic cancer cell lines [12], as well as anti-inflammatory and anti-oxidative effects [13]. Interestingly, soluble and insoluble dietary fibers from kiwifruit regulate gut microbiota to prevent rats from developing T2D triggered by a high-fat diet and streptozotocin (HFD/STZ) [14].

In light of the aforementioned rationale, the fundamental objective of the current research was to understand the overall protective effects of kiwi extract on hyperglycemia brought on by a high-fat diet and streptozotocin in Albino rats. The work looks at the mechanisms by which they affect oxidative stress-mediated inflammation and apoptosis in renal and hepatic tissues. In addition, phosphoinositide 3-kinase (PI3K), protein kinase B (Akt), and mammalian target of rapamycin (mTOR) signaling, which is primarily expressed in insulin-responsive tissues and plays a significant role in intracellular physiology by controlling growth factor signals essential for cellular functions like glucose homeostasis, lipid metabolism, and protein synthesis, were also studied. Moreover, the study investigated the molecular docking and dynamic simulation studies of phytochemical constituents of kiwifruit extract.

## 2. Results

### 2.1. Body Weight Changes after Administration of Kiwi Extract

The mean body weights of the groups were comparable at the start of the experiment. In the control groups, there was a consistent normal body weight gain. An obese rat model was successfully created after 8 weeks of HFD feeding, as seen by the noticeable increase in body weight in HFD-fed groups compared to the control group; furthermore, throughout the entire experiment, a reduction in body weight was noticed after STZ induction. The weekly variations in body weight levels in the control and experimental groups are depicted in Figure 1. Two doses of kiwi extract along with insulin were administered during the treatment period, and this reduced body weight loss throughout the experiment compared to the untreated groups.

### 2.2. Effect of Kiwi Extract Administration on Serum Glucose and Insulin Levels of T2DM-Induced Rats

Table 1 reveals the influence of kiwi treatment on fasting glucose and fasting insulin levels in serum. The T2DM group caused a significant rise, more than four-fold (*p* < 0.01), in glucose levels compared with the normal group. Conversely, the levels of glucose in the T2DM group treated with dose I and II of the kiwi extract exhibited a marked decline (*p* < 0.01) compared to the T2DM group. Furthermore, blood glucose levels in the diabetic group treatment with 200 mg/kg of kiwi extract showed no significant difference with control levels. Meanwhile, treatments with both doses (100 and 200 mg/kg of kiwi extract) were more effective than insulin treatment. Additionally, compared to the normal group, the T2DM group considerably raised insulin levels by upward of 15 times (*p* < 0.01), while treatment with kiwi revealed a noticeable drop in the levels of insulin compared to the T2DM group (*p* < 0.01).

In T2DM, home-collection hemoglobin A1c (HbA1c) levels were significantly elevated (*p* < 0.01) in comparison to the normal control group. HbA1c levels were well regulated in the kiwi-extract-treated diabetic group compared to diabetic rats (*p* < 0.01). The levels of HbA1c were well regulated near to normal levels in the kiwi-extract-treated diabetic group (200 mg/kg).

Based on Table 1, after induction with HFD/STZ, the levels of the insulin resistance index (HOMA-IR) were found to increase in contrast to normal control animals, and they remained high for the entirety of the study (*p* < 0.01).

### 2.3. Effect of Kiwi Extract Administration on Serum AGEs Levels

Prolonged hyperglycemia is connected to an increase in advanced glycation end-product (AGE) levels in the serum of T2DM-induced rats compared to healthy controls (*p* < 0.01). In our investigation, two doses of kiwi extract and insulin administration caused a substantial drop in AGE levels compared to untreated rats. The best results were shown in groups treated with 200 mg/kg kiwi extract, which achieved levels comparable to those of the control group (Figure 2; *p* < 0.01).

### 2.4. Effect of Kiwi Extract Administration on Serum Adipokines Levels

When compared to the control animals, rats that were given HFD/STZ had significantly greater serum leptin levels (*p* < 0.01). As opposed to the control animals, rats that were given FD/STZ had significantly lower serum adiponectin levels (*p* < 0.01). Compared to the untreated group, the two dosages of kiwi extract significantly decreased leptin levels (Figure 3A) and significantly increased adiponectin levels (Figure 3B; *p* < 0.01). Compared to other doses and with values close to the control, groups treated with 200 mg/kg of kiwi extract showed a more powerful effect on serum adipokine levels.

### 2.5. Kiwi Extract Attenuates Inflammation in the Hepatic and Renal Tissues of T2DM-Induced Rats

We then looked at inflammation in the liver and kidney of induced rats to see how kiwi extract works to reduce inflammation (Figure 4). Liver and kidney levels of TNF-α (Figure 4A,B), INF-γ (Figure 4C,D), IL-6 (Figure 4E,F), and NF-κB (Figure 4G,H) were raised in T2DM-induced rats compared to control rats (*p* < 0.01). In contrast, tumor necrosis factor-alpha (TNF-α), interferon-gamma (IFN-γ), interleukin IL-6 (IL-6), and nuclear factor kappa-light-chain-enhancer of activated B cells (NF-κB) levels were shown to be reduced by the administration of kiwi extract at two doses as well as insulin when compared to the untreated group (*p* < 0.01). The 200 mg/kg dose of kiwi extract had a considerably stronger effect than the other dose plus insulin.

### 2.6. Kiwi Extract Promotes the Phosphorylation of the AKT Signaling Pathway in Liver of T2DM-Induced Rats

An important regulator of glucose absorption through glucose transporter type 4 (GLUT4) is the insulin signaling pathway, which comprises AKT, PI3K, and mTOR. In order to interpret the molecular basis of the effect of two dosages of kiwi extract on the liver of diabetic rats, we performed western blotting. The HFD/STZ-induced rats had lower levels of phosphorylated AKT, PI3K, and mTOR than the control rats, as shown in Figure 5. However, this attenuation was alleviated after the administration of both doses of kiwi extract, which caused a significant increase (*p* < 0.01) in the phosphorylation of AKT (Figure 5A), PI3K (Figure 5B), and mTOR (Figure 5C) in liver tissue compared to untreated rats. In addition, the administration of kiwi extract at a dose of 200 mg/kg demonstrated a more noticeable increase in the phosphorylation levels of insulin signaling members compared to the control values, whereas the other dose as well as insulin demonstrated values that were lower than the control.

### 2.7. Effect of Kiwi Extract on an Apoptotic Pathway in the Liver and Kidney of T2DM-Induced Rats

Bcl-2 expression levels in the liver and kidney of T2DM-induced rats were significantly lower under stress than they were in the control group (Figure 6A,B; *p* < 0.01); however, Bax (Figure 6C,D) and casp3 (Figure 6E,F) expression levels were significantly higher. After two doses of the kiwi extract were administered to model rats, the expression of the pro-apoptotic proteins Bcl-2-like protein (Bax) and cysteine-dependent, aspartate-specific peptidase, and caspase-3 (Casp3) were considerably reduced, whereas B cell lymphoma 2 (Bcl-2) expression was elevated.

### 2.8. Evaluation of the Antioxidant Upshot of Kiwi Extract Administration in HFD/STZ-Induced Rats

Hepatic lipid peroxidation (LPO) was significantly (*p* < 0.01) increased by more than six-fold in the diabetic group compared with the control group, while renal LPO was significantly (*p* < 0.01) increased by two-fold in the diabetic group compared with the control group. Rats which received two doses of kiwi extract exhibited a substantial decline in renal and hepatic LPO contents (*p* < 0.01), as shown in Table 2. In addition, diabetic rats had a significant reduction in reduced glutathione (GSH) levels in the kidney and the activities of superoxide dismutase (SOD), catalase (CAT), glutathione peroxidase (GPx), glutathione-S-transferase (GST), and glutathione reductase (GR) compared with the corresponding control values, as shown in Table 2 and Table 3. On the other hand, kiwi treatment could safeguard the antioxidant enzymes in renal tissues against the destructive upshot of HFD/STZ.

Diabetic rats treated with kiwi extract stimulated a significant improvement (*p* < 0.01) in the hepatic GSH levels and the hepatic activities of SOD, CAT, GPx, GST, and GR. Additionally, these results indicate that treatment with 200 mg/kg of kiwi extract for diabetic rats highlighted no significant difference in the antioxidant system in renal and hepatic tissues compared to normal rats, and both doses of kiwi extract were more effective than the insulin treatment (Table 2 and Table 4).

### 2.9. Evaluation of Kiwi Extract Upshot on Serum Lipid Profile in HFD/STZ-Induced Rats

The diabetic rats showed remarkable amplified levels (*p* < 0.01) of total cholesterol, LDL-cholesterol, triglyceride, and total lipid levels in comparison to the control group. In contrast, the diabetic rats had a noticeable decrease (*p* < 0.01) in the level of HDL-cholesterol in comparison to the control group. Diabetic rats treated with kiwi extract ameliorated these changes in the lipid profile compared to the diabetic rats with no noteworthy difference from the normal control, as presented in Table 5.

Treatment with insulin or two doses of kiwi extract also significantly reduced lipid profile markers related to diabetes (*p* < 0.01). In contrast, treatment with insulin or 100 mg/kg of kiwi extract still significantly increased the levels of lipid profile markers when compared with normal control rats (*p* < 0.01). Furthermore, the diabetic group treated with 200 mg/kg of kiwi extract showed a remarkable improvement through reducing the level of lipid profile markers with no significant difference in lipid profile markers with the control levels, and this was more efficient than insulin treatment.

### 2.10. Evaluation of Kiwi Extract Upshot on Serum Renal and Hepatic Functions in HFD/STZ-Induced Rats

As displayed in Table 6, the present results showed a marked increase in serum creatinine, urea, and uric acid levels in HFD/STZ-induced rats when contrasted with normal animals (*p* < 0.01). When two doses of kiwi extract or insulin were given to HFD/STZ-induced rats, the levels of creatinine, urea, and uric acid significantly decreased and were within the normal range compared to the HFD/STZ-untreated group (*p* < 0.01).

Similarly, a substantial elevation was observed in serum alanine aminotransferase (ALT), aspartate aminotransferase (AST), and alkaline phosphatase (ALP) activities in HFD/STZ-induced rats compared to normal rats (*p* < 0.01). A reverse pattern was exhibited when the HFD/STZ-induced group was treated with kiwi extract. The treated groups revealed a significant reduction in serum ALT, AST, and ALP compared to the diabetic untreated group (Table 6; *p* < 0.01).

### 2.11. Histopathological Analysis of the Liver

To study the effect of kiwi extract as well as insulin on histopathological alterations in the liver tissue of T2DM-induced rats, paraffin sections were stained with hematoxylin and eosin (H&E) for morphological changes in rat liver tissues. Figure 7 displays the liver of the control group with normal hepatic cells (black arrows) and a clear central vein (red arrows). On the other hand, the liver of the T2DM-induced group showed a complete loss of the normal architecture with a focal area of hepatic necrosis occupied by mononuclear cell infiltration (blue arrows) and a larger degenerated area, which was occupied by centrilobular congestion and congested dilated portal tracts along with hemorrhage (red arrows). Additionally, all hepatocytes suffered from vacuolar and hydropic degeneration (fatty degeneration) (black arrows). Moreover, the T2DM-plus-kiwi-extract-treated group (100 mg/kg) revealed a loss of the normal architecture with a large area of hepatic necrosis occupied by mononuclear cell infiltration (red arrows), with all hepatocytes suffering from vacuolar and hydropic degeneration (fatty degeneration) (black arrows). The T2DM-plus-kiwi-extract-treated group (200 mg/kg) appeared with normal hepatocytic cords with almost normal hepatic cells (black arrows) with dilated sinusoids in between (blue arrows) and a clear central vein surround by a small area of mononuclear cell infiltration (red arrows). Finally, the liver of the T2DM-plus-insulin-treated group showed loss of the normal architecture with a large area of hepatic necrosis occupied by mononuclear cell infiltration surrounding a clear central vein (red arrows), with most hepatocytes suffering from vacuolar and hydropic degeneration (fatty degeneration) (black arrows).

### 2.12. Molecular Docking

#### 2.12.1. Molecular Docking Illustration Confirms Silent Information Regulator 1 (SIRT-1) and Novel Components of Kiwi Extract Interaction

The binding mode of resveratrol (co-crystalized ligand) showed an energy binding of −6.90 kcal/mol. against the SIRT-1 target site. Resveratrol formed two pi–alkyl interactions with Pro212 and Ala295; on the other hand, resveratrol interacted with Asp292, Asp298, and Lys444 by three hydrogen bonds with bond lengths of 1.70, 1.50, and 1.69 Å, respectively (Figure 8A).

The binding mode of quercetin exhibited an energy binding of −6.50 kcal/mol. against the SIRT-1 target site. The binding was formed of six hydrophobic interactions with Pro212, Ala295, Phe414, Val445, and Gln294. Additionally, quercetin formed three hydrogen bonds with Asp292, Asp298, and Lys444 with bond lengths of 1.95, 1.90, and 1.93 Å, respectively (Figure 8B,E).

The binding mode of chlorogenic acid exhibited an energy binding of −6.04 kcal/mol. against the SIRT-1 target site. The binding was formed of three hydrophobic interactions with Pro212, Ala295, and Gln294. Moreover, chlorogenic acid interacted with Asp292, Ile210, Arg446, and Lys444 by four hydrogen bonds with bond lengths of 1.89, 2.43, 2.35, and 2.15 Å, respectively (Figure 8C,E).

The binding mode of melezitose exhibited an energy binding of −7.65 kcal/mol. against the SIRT-1 target site. Melezitose formed four hydrogen bonds with Glu208, Arg446, and Lys444 with bond lengths of 2.15, 2.54, 2.83, and 2.24 Å, respectively (Figure 8D,E).

#### 2.12.2. Molecular Docking Illustration Confirms Phosphatase and Tensin Homolog (PTEN) Inhibition and Novel Components of Kiwi Extract Interaction

The binding mode of syringic acid exhibited an energy binding of −6.07 kcal/mol. against the PTEN target site. The binding was formed of three hydrophobic interactions with Ala126, Asp92, and Lys128, with one forming an ionic interaction with Arg130. Additionally, syringic acid interacted with Thr131, Lys125, Gly127, and His93 by four hydrogen bonds with bond lengths of 4.31, 1.95, 2.39, and 2.27 Å, respectively (Figure 9A,E).

The binding mode of p-coumaric acid exhibited an energy binding of −5.98 kcal/mol. against the PTEN target site. Moreover, p-Coumaric acid formed two hydrophobic interactions with His93 and Lys128, with one forming two ionic interactions with Arg130. Additionally, p-coumaric acid interacted with Thr131, Lys125, Gly127, and Ala126 by four hydrogen bonds with bond lengths of 2.32, 1.89, 2.22, and 2.95 Å, respectively (Figure 9B,E).

The binding mode of caffeic acid exhibited an energy binding of −6.06 kcal/mol. against the PTEN target site. It was formed two hydrophobic interactions with His93 and Lys128, with one forming an ionic interaction with Arg130. Additionally, caffeic acid formed four hydrogen bonds with Thr131, Lys125, Gly127, and Gln171 with bond lengths of 4.01, 1.83, 2.60, and 2.33 Å, respectively (Figure 9C,E).

The binding mode of ferulic acid exhibited an energy binding of −6.15 kcal/mol. against the PTEN target site. Ferulic acid formed three hydrophobic interactions with Ala126, His93, and Lys128. Additionally, it interacted with Thr131, Lys125, Arg30, and Gly127 by four hydrogen bonds with bond lengths of 3.98, 1.87, 2.90, and 2.38 Å, respectively (Figure 9D,E). 

Following the completion of the molecular docking, the top 20 poses were taken; the docking (affinity) score (DG) and root mean square deviation (RMSD) values are compiled in Table 7.

#### 2.12.3. Molecular Dynamic (MD) Simulation Studies

##### Protein and Ligand RMSD Analysis

Two compounds were selected for MD simulations, including quercetin complexed with SIRT-1 and syringic acid complexed with PTEN. The conformational stability of the proteins was monitored through the C*α* atoms of the protein with respect to their initial position. As shown in Figure 10A,B, the syringic acid PTEN complex showed high stability, with an RMSD value within 3.00 Å, compared to the quercetin SIRT-1 complex, which is acceptable for such proteins.

The RMSD was measured for ligands’ atoms based on the initial position of the heavy atoms inside the protein active site. The syringic acid PTEN complex showed stability over time at the active site, as shown in Figure 10A,C; however, the quercetin SIRT-1 complex showed some fluctuation at amino acids region 40–60 at approximately 30–70 ns and then showed stability later, as shown in Figure 10B. The SIRT-1 protein showed high numbers of conformational changes over the MD simulation period that exhibited more fluctuations until 70 ns of the simulation, as shown in Figure 10D. It showed less stability within the active site, as shown in Figure 10B.

On the other hand, the compactness of the complex was represented by the radius of gyration (Rg). A lower the degree of fluctuation throughout the simulation period indicates a greater compactness of a system. The Rg of the syringic acid PTEN complex was found to be less than the starting period, while the quercetin SIRT-1 complex was found to be higher than the starting period. Interactions between protein–ligand complexes and solvents were measured by the solvent-accessible surface area (SASA) over the simulation period. Therefore, the SASA of the complex was calculated to analyze the extent of the conformational changes that occurred during the interaction. Interestingly, the syringic acid PTEN complex featured a reduction in the surface area showing a relatively low SASA value than the starting period compared to the quercetin SIRT-1 complex, as shown in Figure 11A,B.

##### Histogram and Heat Map Analysis

Since syringic acid showed the highest stability within the PTEN active pocket, their interactions are discussed in detail. Syringic acid formed H-bond interactions with the following residues: Lys125 (~100%), Ala126 (~100%), Gly127 (~100%), Lys128 (~80%), Arg130 (~90%), Thr131 (~200%), and Cys124 (~10%), as presented in Figure 12A. Whereas quercetin showed moderate stability within the SIRT-1 active pocket, their interactions are discussed in detail. Quercetin formed H-bond interactions with the following residues: Arg197 (~3%), Lys203 (~3%), Leu206 (~45%), Pro207 (~7%), Glu208 (~15%), Thr209 (~15%), Ile210 (~5%), Pro210 (~13%), and Asp228 (~25%), as presented in Figure 12B.

Another type of H-bond interaction is the water-bridged H-bond, where crystal water molecules form a link between the protein residues and ligands. Syringic acid formed water-bridged H-bonds with residues Asp92 (~10%), His93 (~10%), Lys128 (~12%), and Arg135 (~35%). Quercetin was able to form more than 30 water-bridged H-bonds interactions, where bridge H-bonds with amino acids (as shown in Figure 12B) held interactions for more than 30% during the simulation time. Along with H-bonds interactions, syringic acid was able to form hydrophobic interactions with residues Val45 (~10%), Cys124 (~5%), Lys128 (~35%), Arg130 (~20%), and Ile168 (~7%) and an ionic interaction with the Arg130 residue. On the other hand, quercetin showed notable hydrophobic interactions with Lys203, Ile210, Pro211, Pro212, Pro213, Leu215, and Phe414. Another method used to monitor these interactions involves plotting the number of interactions with respect to time; a heat map (Figure 13A,B) indicates the number of interactions at each frame of the syringic acid PTEN and quercetin SIRT-1 complexes, whereas the dark color indicates more interactions. From the heat maps, it was observed that the highest number of conformations of the protein formed up to nine hydrogen bonds of the syringic acid PTEN complex and five for the quercetin SIRT-1 complex.

##### MM-GBSA Calculations

The molecular mechanics, with a generalized Born and surface area solvation (MM–GBSA), were carried out to calculate both the ligand binding strain and free energies for docked ligands over the 100 ns. The ΔG binding energies, Coulomb energies, hydrogen bond energies, covalent binding energies, lipophilic energies, and van der Waals energies were recorded. The obtained results are described in Figure 14A,B in more detail.

As shown in Figure 14A,B, the syringic acid PTEN complex showed high MM-GBSA binding energies with almost −30 kcal/mol, while the quercetin SIRT-1 complex showed moderate MM-GBSA binding energies with almost −9.71 kcal/mol. In addition, the syringic acid PTEN complex showed higher coulomb energy and higher H-bond energy compared to the quercetin SIRT-1 complex.

## 3. Discussion

T2DM has become a serious global health concern in recent decades due to the epidemic’s increasing incidence [1]. In the current study, the T2DM standard model triggered by HFD/STZ was implemented to examine the anti-diabetic, hypolipidemic, and antioxidant efficiency of *Actinidia deliciosa* (kiwi) extract. Using this commonly applied model, the pathophysiology of T2DM was scrutinized. These findings confirmed a previous study performed by Goyal et al. [15], which showed that STZ inhibited glucose metabolism, which was put forth when the authors declared that the HFD administration promptly followed by the injection of a tiny dose of STZ induced insulin resistance along with the development of hyperglycemia, hyperinsulinemia, and dyslipidemia in the experimental rats.

One of the principal hallmarks of T2DM is insulin resistance (IR), which manifests as decreased tissue insulin sensitivity. Administration of HFD for 2 weeks has been shown to only mildly increase insulin resistance and/or glucose intolerance, but rats fed a high-fat diet for 5 weeks consecutively demonstrated greater body fat percentages, diminished glucose tolerance, and hyperinsulinemia [16]. The HFD/STZ rat paradigm is a good choice for assessing IR since STZ treatment to HFD-fed animals resulted in severe hyperglycemia. The development of therapeutic drugs for T2DM must therefore focus on IR [17]. According to our insights, controlled diabetic obese rats had considerably elevated fasting blood glucose and insulin levels as well as a more a pronounced insulin resistance index. Kiwi extract administration dramatically improved glycemic management by dropping fasting glucose and insulin levels simultaneously with an actual improvement in insulin sensitivity.

Based on the above, elevated blood glucose levels trigger tissue damage through a variety of processes, incorporating the buildup of AGEs in the blood, which may serve as an informative sign of diabetic complications. AGEs are heterogeneous substances, created when extra sugars combine with the amino groups in proteins and nucleic acids. Increased AGEs are linked to poor glycemic management and insulin resistance [17]. Our insights highlighted a considerable rise in serum AGE levels, which was effectively reduced by both doses of kiwi extract treatment.

By organizing energy and inflammation, adipocytokines, which are extensively expressed in adipose tissue, support homeostasis. Adipocytokine fluctuations are thought to be a symptom of adipose tissue dysfunction. Adipocytokines could also provide pertinent details on the pathophysiological origins of T2DM [18]. Leptin and adiponectin hormones generated from adipocytes play an indispensable role in the etiology of T2DM through heightened insulin sensitivity [17]. Under typical circumstances, serum leptin levels are low while adiponectin is raised. These adipocytokines are noteworthy markers that are essential for sustaining the harmony between lipid and carbohydrate metabolism, which, over time, regulates inflammation and insulin signaling [19].

Any variation in these adipocytokines’ levels points to adipose tissue dysfunction and the consequent activation of pathways that eventually result in insulin resistance. Leptin is accountable for bolstering insulin resistance through various means. Adiponectin increases insulin sensitivity in the liver by suppressing hepatic glycogenesis [20]. When compared to the control group, blood leptin levels in the T2DM group were considerably higher while adiponectin levels were lower. Insulin resistance, which consequently results in type 2 diabetes, is triggered by the irregularity of these inflammatory markers [17]. In comparison to the insulin-treated group, kiwi extract treatment ameliorated the levels of these adipocytokines to levels that were nearly normal. In a similar line, Rebecca et al. [18] documented that an ethanolic extract of *Carica papaya* enhanced insulin resistance in high HFD/STZ-induced T2DM with an increase in adiponectin and a drop in leptin synthesis in fat cells. Likewise, this also occurs in rat models with high-fat diets and streptozotocin induction.

Based on studying the molecular pathways behind insulin resistance that occurred in rats with HFD/STZ-induced T2DM, the anti-diabetic efficacy of kiwi extract was investigated. The insulin signaling pathway, which entails AKT, PI3K, and mTOR, is an enormous regulator of glucose absorption through GLUT4 [21]. Dysregulation is a major contributor to the glycemic response seen in uncontrolled diabetes, as the insulin signaling pathway regulates the transport of glucose in hepatic cells. Previous research reported that boosting p-IRS1 and p-AKT might promote glucose absorption and minimize blood glucose levels [22].

According to HFD/STZ treatment, levels of phospho-insulin receptor substrate-1 (p-IRS1), phospho-phosphoinositide 3-kinase (p-PI3K), and phospho-protein kinase B (p-AKT) were shown to be substantially diminished [17]. Previous research has highlighted that boosting p-IRS1 and p-AKT could enhance glucose absorption and lower blood glucose levels [17,23]. In a comparable manner, Shokouh et al. [24] documented that the expression of the mTOR gene was downregulated in the coffee groups, resulting in improved insulin sensitivity in a T2DM hepatic rat model. Particularly high levels of circulating amino acids and increased fat have been shown to contribute to nutritional overload, which boosts mTOR activation and can trigger insulin resistance in peripheral insulin-responsive tissues. The results of the present research revealed that oral treatment of kiwi extract dramatically enhanced the protein levels of p-PI3K and p-AKT in liver cells. However, kiwi extract had the strongest impact in restoring these protein levels when compared to insulin treatment.

Likewise, obesity typically throws off the inflammation balance, which raises the concentration of proinflammatory cytokines. Peripheral insulin resistance and hyperglycemia can be ignited by inflammation [18]. We focused on the protein levels of proinflammatory cytokines such as NF-κB, TNF-α, INF-γ, and IL-6 in both control and experimental rats to ascertain whether kiwi extract can lessen inflammation and insulin resistance brought on by HFD and STZ. In our study, the T2DM-induced group had elevated protein levels of NF-κB, TNF-α, INF-γ, and IL-6, depicting increased inflammation with regard to the control group. The treatment with kiwi extract lowered NF-κB, TNF-α, INF-γ, and IL-6 more effectively than that of the insulin treatment. This exhibits the potential for kiwi extract to suppress the formation and secretion of pro-inflammatory cytokines. Similarly, adiponectin behaves as an anti-inflammatory cytokine, which is incredibly important in the pathogenesis of T2DM. Kiwi extract could diminish the inflammation by downregulating other inflammatory cytokines, such as TNF-α, leptin, and upsurges in adiponectin levels.

Additionally, recent research has shown that tissue damage, apoptosis, and consequent organ failure can result from the oxidative stress and inflammation that hyperglycemia causes [4,25]. Consequently, suppressing the corresponding rise in oxidative stress and inflammation is one of the most crucial approaches to avoiding programmed cell death in diabetes [26]. The current study proved that administering rats with HFD/STZ drove apoptosis. An anti-apoptotic protein, Bcl-2, adheres to the mitochondrial membrane and prevents cellular death by restricting the release of cytochrome-C [27]. The pro-apoptotic molecule Bax, on the other hand, is present in healthy cells but is dormant. In response to apoptotic stimuli, Bax succumbs to conformational activation, which impairs Bcl-2’s ability to inhibit apoptosis [28,29].

In the current study, we assessed the relative gene expression of the anti-apoptotic protein, Bcl-2, and the pro-apoptotic proteins, Bax and casp-3, in the experimental groups. Bax and caspase-3 cleavage levels were dramatically boosted but the antiapoptotic protein Bcl-2 was drastically reduced in the HFD/STZ group. Kiwi extract treatment exerted anti-apoptotic activity through upregulating Bcl-2 and downregulating Bax and casp3 compared to the groups with diabetes.

Oxidative stress is heightened by an excess of ROS and a deficiency in antioxidant mechanisms, and it has a crucial role in the inception and perpetuation of complications in diabetes [30,31]. ROS generation is triggered by T2DM-induced dyslipidemia and oxidative stress, subsequently leading to escalated LPO and membrane damage [32]. In our study, the LPO levels in the T2DM group rose sharply, and a significant considerable drop in the activity of GSH, SOD, CAT, GPx, and GST was observed, signifying an explicit state of oxidative stress. In related research by Othman et al. [33], flavonoids from *Ocimum basilicum* extract were shown to reduce oxidative stress indicators in HFD-STZ diabetic rats. The removal of free radicals from cells and the amelioration of oxidative stress are achieved by enzyme-based antioxidants, notably SOD, GPx, and CAT. A decline in these antioxidant enzymes triggers insulin resistance, culminating in diabetes mellitus. When compared to the control group in the current study, the enzymatic antioxidants CAT, SOD, GPx, and GSH were noticeably decreased in the T2DM rats [25]. Treatment with kiwi extract greatly boosted antioxidant enzyme levels, considerably reducing ROS and inhibiting lipid peroxidation in T2DM more efficiently than treatment with insulin. These results were rather similar to those of a study of Aleid et al. [34], as kiwi extract upregulated SOD, CAT, GSH, and GPx levels and mitigated lipid peroxidation against indomethacin-induced oxidative stress.

Based on the above, the biochemical and molecular evaluations were supported by the histopathology findings. In HFD/STZ-induced rats, the liver tissue microarchitecture was seriously disrupted. The results of the histopathology evaluation highlight that kiwi extract has ameliorating effects on liver tissue consequent to the repair of cell architecture [25].

Moreover, the molecular docking results suggest that the tested metabolites exhibit favorable binding energies and form specific interactions with the target sites of SIRT-1 and PTEN. The binding energies indicate the stability of the ligand–protein complexes, with lower values indicating stronger binding.

According to SIRT-1 target site binding domain, resveratrol, quercetin, chlorogenic acid, and melezitose all displayed significant binding energies, suggesting their potential as effective natural products for activating the SIRT-1 pathway. These ligands formed a combination of hydrophobic interactions and hydrogen bonds with key residues, such as Pro212, Ala295, Asp292, Asp298, and Lys444, crucial for the stabilization and specificity of ligand binding.

Regarding the PTEN target site, syringic acid, p-coumaric acid, caffeic acid, and ferulic acid demonstrated favorable binding energies with strong binding interactions that indicates the ability of these phenolic acids to inhibit the PTEN active site. These ligands formed hydrophobic interactions with important residues like Ala126, His93, and Lys128. Additionally, they engaged in hydrogen bonding interactions with Thr131, Lys125, Gly127, and other key residues. The presence of ionic interactions, such as those observed with Arg130, further contributes to ligand binding and stability.

Overall, the results of our molecular docking study suggest that resveratrol, quercetin, chlorogenic acid, melezitose, syringic acid, p-coumaric acid, caffeic acid, and ferulic acid possess the potential as ligands for targeting SIRT-1 and PTEN. Their favorable binding energies and specific interactions with key residues within the binding sites support their further investigation as potential therapeutic agents. However, it is important to note that additional experimental studies, such as in vitro and in vivo evaluations, are necessary to validate the binding affinities and biological activities of these ligands and assess their potential as drug candidates. A schematic illustration of the effects of kiwi extract on the different pathways is shown in Figure 15.

## 4. Materials and Methods

### 4.1. Preparation of Aqueous Extract of Kiwifruit

In January 2022, we purchased *Actinidia deliciosa* (kiwifruit) from a near market. The whole fruit was homogenized with a high-powered hand mixer. The ethanol extract was made by soaking the flesh in 70% ethanol (1:1 *w*/*v*) for 24 h at 30 °C. Subsequently, the filtrates were concentrated using a rotary vacuum evaporator (RE-110EX, Labfirst Scientific Instruments (Shanghai) Co., Ltd., Shanghai, China), and thereafter lyophilized with a Freeze Dryer (STELLAR^®^ Laboratory Freeze Dryer, Millrock Technology, Inc., Kingston, NY, USA) and preserved at −80 °C until use. The needed amount of kiwifruit extract powder was dissolved in distilled water throughout the study [12].

### 4.2. Animals and the Experimental Design

All experimental procedures performed were approved be the Committee of the Faculty of Science, Alexandria University, Egypt (AU 04221126302), where experiments were conducted in keeping with the regulations provided by Alexandria University’s Animal Protocol Guidelines.

#### 4.2.1. Experimental Design

Thirty male Albino rats (220–250 g) were selected from the Experimental Animal Holding of the National Research Centre, Giza, Egypt. The rats were placed in normal polypropylene cages (each cage contained three rats) and kept in a controlled room temperature and humidity cycle for 12/12 h. During the first week, all rats were fed a standard diet with a total energy of 3.45 kcal/g (26% protein, 3% fat, 54% carbohydrate and 17% vitamins and minerals percentage per 100 gm).

Six rats completed the experiment with a standard diet and water, considered as the normal group (first group, NC), while 24 of rats were fed a high-fat diet (HFD) (total energy 44.3 kJ/kg containing 22% fat, 48% carbohydrate, and 20% protein) for 8 weeks [17]. After fasting overnight, the rats obtained an intraperitoneal streptozotocin (STZ) (35 mg/kg) injection, freshly prepared in a citrate buffer (0.05 M, pH 4.5) [35]. The glucose level the in the blood was tracked three days later using ACCU CHEK PERFORMA (Roche Diagnostics, Mumbai, India) blood glucose meters. The experiment involved rats with blood glucose levels >300 mg/dL.

The diabetic rats were initially randomly divided into four sets, with six rats in every group. The second group (diabetic group, DC) was fed a HFD for eight weeks. In the third group, the rats received sub-cutaneous injections of insulin (2 U/rat) once daily for an 8-week period [36]. Group four (DKI) was the diabetic group and obtained 100 mg/kg of body weight of kiwi extract orally per day as a treatment. In group five (DKII), the rats were treated with 200 mg/kg of body weight of kiwi extract orally by an orogastric gavage technique every day. The administration of doses occurred consistently at a fixed time each day following breakfast. The kiwi extract solutions were produced by dissolving the appropriate dosage of lyophilized powder, equivalent to either 100 mg or 200 mg of extract, in distilled water immediately before use. The amount of extract used for each rat was determined based on their body weight. Subsequently, the rats consumed the prepared solutions via oral gavage.

#### 4.2.2. Evaluation of Body Weight Changes

Weekly reports of each rat’s body weight in each group were performed throughout the course of the study in order to measure body weight changes.

#### 4.2.3. Biological Samples

Over the subsequent 8 weeks, the rats were fasted overnight and then subjected to anaesthesia by the nose cone method with isoflurane (30% *v*/*v* isoflurane in propylene glycol). Then, the blood was extracted by cardiac puncture and centrifuged for 10 min at 2325× *g* using an LST-O-1 Centrifuge (CGOLDENWALL Manufacturer, Manchester, UK). The serum was stored at −80 °C until the assays were complete. The renal and hepatic tissues were removed and washed in cold 0.9 N NaCl and preserved at −80 °C for biochemical analysis.

### 4.3. Biochemical Marker Estimation in Blood and Tissues

#### 4.3.1. The Diabetic Markers

Fasting glucose was evaluated in serum by a colorimetric method according to Buzanovskii [37]. Additionally, the HbA1c was assessed by standard colorimetric kits (BioAssay Systems, Hayward, CA, USA) according to the manufacturer’s guidelines. Fasting insulin was quantified by a quantitative ELISA kit quantified based on the method of Matthews et al. [38]. In addition, the homeostasis model of assessment for HOMA-IR was calculated using the following formula: HOMA-IR = fasting insulin (IU mg/L) × fasting glucose (mg/dL)/405.

#### 4.3.2. Assessment of Diabetogenic-Related Parameters in the Serum of Rats

Leptin (MyBioSource, San Diego, CA, USA, MBS012834), adiponectin (MyBio-Source, MBS068220), and AGE (MyBio-Source, MBS261131) concentrations were analyzed by sandwich immunoassays according to the manufacturer’s instructions in the serum of rats. Serum was collected and centrifuged at 1000× *g* for approximately 20 min. The supernatant was collected carefully and assayed immediately. The experimental setup involved the utilization of a plate reader (Agilent Technologies, Inc. Headquarters, located in Santa Clara, CA, USA). Optical densities were measured at a wavelength of 450 nm. The optical density of the blank was subtracted from each standard and sample in order to calculate the readings. Subsequently, professional curve fitting software (Agilent GPC data analysis software, B.01.01) was employed to generate standard curves, which exhibited a linear relationship. These curves were then utilized to determine the levels of each analyte.

#### 4.3.3. ELISA Iimmunoassay for Quantification of Inflammatory Parameters

Frozen kidney and liver tissues were homogenized in a lysis buffer (150 mM NaCl, 10 mM Tris solution, 1% Triton X-100, pH 7.4) including protease inhibitors, subsequently centrifuged at 22,673× *g* at 4 °C for 10 min, and the supernatant was collected. Then, the total protein content of the renal and hepatic homogenates was evaluated by a formerly informed protein assay [39]. For the assessment of IL-6, IFN-γ, and TNF-α in the liver and renal tissues of all the studied groups, the commercially rat-specific ELISA kit (Abcam, Cambridge, UK; ab236712, ab46107, and ab100785, respectively) have been specifically developed for the purpose of accurately quantifying the concentration of these target proteins. The ELISA kits utilize a capture antibody that is labelled with an affinity tag, as well as a detection antibody that is coupled with a reporter. These antibodies work together to immunocapture the analyte present in the solution. The complete assemblage consisting of the capture antibody, analyte, and detector antibody is subsequently immobilized by the immunoaffinity of an anti-tag antibody that coats the well. In order to conduct the experiment, the wells were first filled with either samples or standards, after which the antibody mix was introduced. Following the incubation period, the wells underwent a washing process to eliminate any material that had not adhered to the surface. The TMB development solution was introduced and incubated, whereupon it was catalyzed by HRP, resulting in the production of a blue color. The reaction was subsequently halted by the introduction of a stop solution, thereby finalizing any alteration in color from blue to yellow. The signal was produced in direct proportion to the quantity of the bound analyte, and its strength was then measured at a wavelength of 450 nm using a plate reader (Agilent Technologies, Inc. Headquarters, Santa Clara, CA, USA). Subsequently, professional curve fitting software was employed to generate standard curves, which exhibited a linear relationship. These curves were then utilized to determine the levels of each analyte. Each sample was double-checked, and the average was reported.

#### 4.3.4. Nuclear Protein Isolation and Assays for Renal and Hepatic NF-κB

According to Dignam et al. [40], nuclear proteins were extracted from renal and hepatic tissue homogenates, and the supernatants (nuclear proteins) were aliquoted and kept at −80 °C. In brief, the livers and kidneys of the experimental animals were isolated at the end of the experimental phase, washed in ice-cold buffer (10 mM Tris-HCL, pH 7.4), and blotted on absorbent paper. Samples were then trypsinized and gently minced using an automatic homogenizer (Ultra-Turrax. Wilmington, NC, USA) to isolate hepatic and renal cells. Cytosolic and nuclear fraction extraction were performed by lysing the cell membrane with a suitable hypotonic lysis buffer containing a protease inhibitor cocktail and tributylphosphine as a reducing agent. The supernatant fractions contained the nuclear extract. After determination of the protein concentration, the prepared extracts were stored in aliquots at −80 °C for subsequent NF-κB assays. After this, the estimation of NF-κB was conducted using an NF-κB p50/65 Transcription Factor kit (EZ-TFA, Universal Transcription Factor Assay, Colorimetric) Catalog #70-500).

#### 4.3.5. Immunoblotting

Frozen liver tissue was homogenized using a special lysis solution (100 mM EDTA, 100 mM NaCl, 0.5% Na-deoxycholate, Nonidet p-40, 10 mM Tris, pH 7.5 with protease inhibitors). Supernatants were collected after samples were centrifuged at 12,000 rpm for 20 min at 4 °C, and the protein contents were calculated using the Bradford assay [39]. The liver samples (50 µg) were boiled in the sample buffer for SDS-PAGE, separated on SDS-PAGE gels, and then transferred onto a nitrocellulose membrane. Blots were pre-incubated with primary antibodies for phospho-Akt (Ser473) (4060), Akt (9272), PI3K p85 (4292), mTOR (2972), phospho-mTOR (5536), and β-actin (4967) (Cell Signaling Technology, Beverly, MA, USA) for 2 h at room temperature before being washed with TBST buffer and incubated overnight at 4 °C. The blots were washed and then incubated with the secondary antibody for 2 h at room temperature. The bands were then quantified using Image Lab 6.1 software after being visualized using an alkaline phosphatase-specific kit (ab83369) [41].

#### 4.3.6. Quantitative Real-Time PCR Analysis

Total RNA was obtained from freshly isolated renal and liver tissues by QIAzol Lysis Reagent following the manufacturer’s instructions (Qiagen, Hilden, Germany, 79306). RNA concentration was quantified via nanodrop, and then reverse-transcribed into cDNA with the Maxima H Minus Reverse Transcriptase following to the manufacturer’s protocol. Bcl-2, Bax, and Casp3, mRNA levels were evaluated using 20 µL amplification mixtures with the QuantiTect SYBR Green PCR Kit containing cDNA and 0.5 M of each primer. The real-time PCR (PCR-RT-96 Series, Bioevopeak Company, Jinan, China) cycling settings were 10 min at 95 °C for polymerase activation, 40 cycles at 95 °C for 10 s, 58 °C for 15 s, and, subsequently, 72 °C for 15 s. The relative expression of mRNA was quantified by the 2^−ΔΔCt^ approach, which was standardized to the mRNA levels of the Actb housekeeping gene [42]. Primer sequence: Bcl-2 (NM_016993.2), F:5′-ATGTGTGTGGAGACCGTCAA-3′ and R:5′-GCCGTACAGTTCCACAAAGGG-3′; Bax (NM_017059.2), F:5′-ATGGAGCTGCAGAGGATGATT-3′ and R:5′-TGAAGTTGCCATCAGCAAACA-3′; caspase-3 (NM_012922.2), F:5′-AATTCAAGGGACGGGTCATG-3′ and R:5′-GCTTGTGCGCGTACAGTTTC-3′; GAPDH (NM_001394060.2), F:5′-GGCACAGTCAAGGCTGAGAATG-3′ and R:5′-ATGGTGGTGAAGACGCCAGTA-3′ [43].

#### 4.3.7. The Antioxidant Markers of the Kidney and Liver Tissues

The kidney and liver tissues were homogenized in cold 50 mmol sodium phosphate buffer, pH 7.2 (1:10 *w*/*v*), and then centrifuged using a High-Speed Refrigerated Centrifuge, benchtop, CFGR-B (Bioevopeak Company, China) for 10 min at 1207× *g*. The supernatant was separated and preserved at −80 °C prior to biochemical analysis. Ellman’s reagent was used to measure the renal and hepatic GSH concentration. The produced golden color as a result of the reduction of DTNB by GSH was recorded at 412 nm immediately, and the GSH level was expressed as ng/mg of protein [44]. Antioxidant enzyme activities, including CAT, at the breakdown of H_2_O_2_ was conducted was quantified by the sample absorbance at a wavelength of 240 nm, and the enzyme activity was determined as nmol of H_2_O_2_ released/min/mg of protein [45]. GR activity measurements, using a methodology that relies on the observed rise in absorbance at a wavelength of 412 nm, were quantified as a result of the reduction of DTNB by GSH. The outcomes were quantified as micrograms of glutathione disulfide (GSSG) used each minute per mg of protein [46]. GPx was measured by the sample absorbance measurements recorded at a wavelength of 412 nm, and the GPx activity was expressed as µmoles of GSH oxidized/min/mg of protein [47]. GST, concisely, was measured by the absorbance for the sample at a wavelength of 310 nm, with a comparison was made against air, and GST activity was expressed as µmoles of CDNB utilized/min/mg of protein [48]. SOD was also assessed using the procedures of Gao et al., and the activity of the SOD in the samples was quantified and expressed in U/mg of protein [49]. In addition, the Janero [50] method was used to calculate the LPO. Briefly, the level of LPO was assessed using the chemical reaction between malondialdehyde (MDA) and thiobarbituric acid (TBA). MDA is a molecule that is produced as a result of membrane lipid peroxidation. The methodology relies on the change in absorbance at a wavelength of 532 nm. The findings were quantified in units of nmol/mg of protein.

#### 4.3.8. The Markers of the Lipid Profile

Lipid profiles were tested for total cholesterol, LDL–cholesterol, HDL-cholesterol, triglycerides, and total lipid, as stated by the manufacturer’s guidelines.

#### 4.3.9. Biochemical Assessment of Serum Biomarkers of the Kidney and Liver Functions

Liver enzyme markers, such as ALP, ALT, and AST, as well as kidney biomarkers, creatinine, urea levels, and uric acid, were evaluated by standard colorimetric kits (BioAssay Systems, Hayward, CA, USA) according to the manufacturer’s guidelines.

### 4.4. Histopathology Study

Liver tissues from all groups were fixed in 10% formalin, then dehydrated using alcohol, and fixed in paraffin. Using a rotatory microtome, sections were cut (3 µm thick) and mounted on clean glass slides. After this, samples were stained using hematoxylin and eosin (H&E) and investigated under a light microscope [51].

### 4.5. Method of Molecular Docking

#### 4.5.1. Preparation of Protein

In the drug design and drug discovery phases, the computational (in silico) technique has been widely used as an efficient instrument for virtual biological screening. This method evaluates the biological activities and estimated affinities of natural products. Several recent applications of computational chemistry have revealed a better understanding of the nature of targeted sites and the identification of different compounds as inhibitors or activators. Using computer-based chemistry approaches was appropriate when evaluating metabolites against the SIRT-1 and PTEN target sites. At first, the target protein was downloaded from the protein data bank (protein Id: 5BTR and 5BZX). The protein and the tested compounds were prepared, and the energy was minimized by an MMFF94 force field. The molecular docking was performed, and 20 poses were generated; then, the best orientations, affinity scores, and RMSD values were captured, as shown in Table 7.

#### 4.5.2. Ligand Preparation

Based on a literature review of the compounds isolated from *Actinidia deliciosa*, seven compounds were selected for docking studies: three compounds (quercetin, chlorogenic acid, and melezitose) were tested against SIRT-1, and four compounds (syringic acid, p-coumaric acid, caffeic acid, and ferulic acid) were tested against PTEN target sites. The 2D structures of these compounds were downloaded in the SDF (Spatial Data File) format from the PubChem database and then converted into the 3D PDB format using an online smiles translator.

#### 4.5.3. Molecular Dynamic (MD) Simulation Studies

To study of the compounds’ stability with the best docking score at the SIRT-1 and PTEN active sites, MD simulations were conducted for 100 ns. The obtained RMSDs for the complexes and the ligands concerning their original positions within the active site were reported and analyzed. Boundary compound interactions were also tested and evaluated in detail. Finally, the MM-GBSA free binding energy was estimated for all complexes during the simulation trajectories [52].

### 4.6. Statistical Analysis

The results are shown as the mean value ± the standard error (SEM). The significant differences among the values were analyzed by one-way analysis of variance analysis followed by Tukey post hoc studies to evaluate the difference between multiple groups. Statistical studies were carried out using IBM SPSS statistics 25 software (International Business Machines Corporation, IBM, Armonk, NY, USA); statistical significance was set at *p* < 0.01.

## 5. Conclusions

*Actinidia deliciosa* (kiwi) extract, according to the current study, showed anti-inflammatory and antihyperglycemic activities by modulating inflammatory molecules such as TNF-α, INF-γ, and IL-6, as well as mTOR, while restoring normal levels of antioxidant enzymes, oxidative stress, and apoptotic markers in type 2 diabetes experimental rats fed a high-fat diet. In addition, we have also proposed, via in silico study, quercetin, chlorogenic acid, and melezitose as components of kiwi extract that docked well with the potential as effective natural products for activating the SIRT-1 pathway. Furthermore, phenolic acids of kiwi extract, especially syringic acid, p-coumaric acid, caffeic acid, and ferulic acid, have a great ability to inhibit the PTEN active site. The current work shows, for the first time, that kiwi extract prevents obesity-induced insulin resistance through controlling insulin signaling pathway parameters such as PI3K, AKT, and mTOR. Therefore, kiwi extract holds immense promise for investigation in clinical studies aimed at finding effective and safe medications for the treatment of type 2 diabetes.

## Figures and Tables

**Figure 1 ijms-24-13759-f001:**
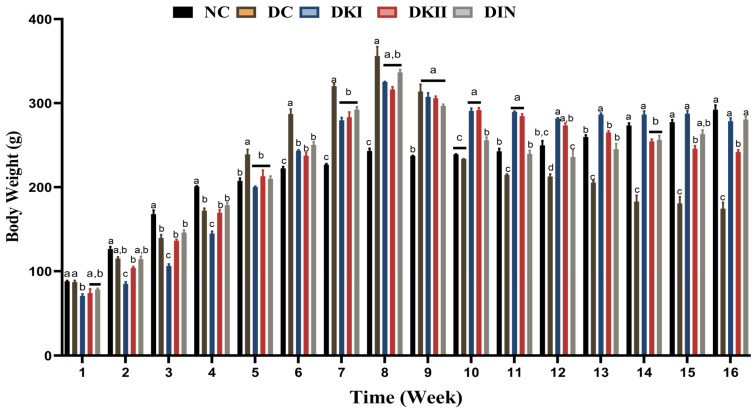
Body weight values of all studied groups expressed in grams. Values are expressed as mean ± SEM (*n* = 6). The means with different letters in each bar (a–d) are significantly different (*p* < 0.01), where the largest data values take the letter (a), and the smallest data values take the letter (d). NC: normal control rats, DC: diabetic control group, C group, DKI: DC group treated with 100 mg kiwi extract, DKII: DC group treated with 200 mg/kg kiwi extract, DIN: DC group treated with insulin.

**Figure 2 ijms-24-13759-f002:**
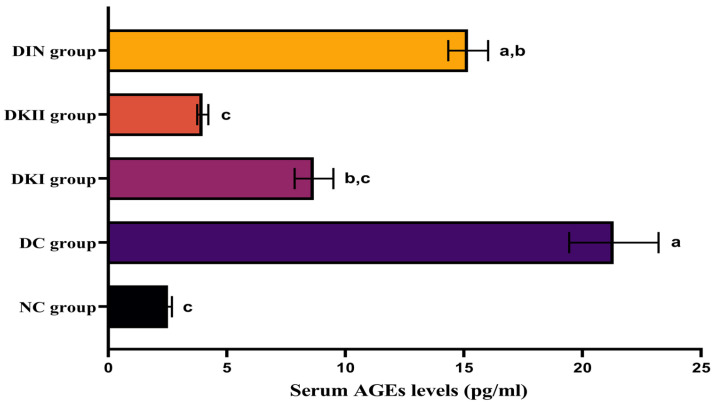
Serum values of AGEs in all the studied groups. Values are expressed as mean ± SEM (*n* = 6). The means with different letters in each bar (a–c) are significantly different (*p* < 0.01), where the largest data values take the letter (a), and the smallest data values take the letter (c). NC: normal control rats, DC: diabetic control group, DKI: DC group treated with 100 mg kiwi extract, DKII: DC group treated with 200 mg/kg kiwi extract, DIN: DC group treated with insulin.

**Figure 3 ijms-24-13759-f003:**
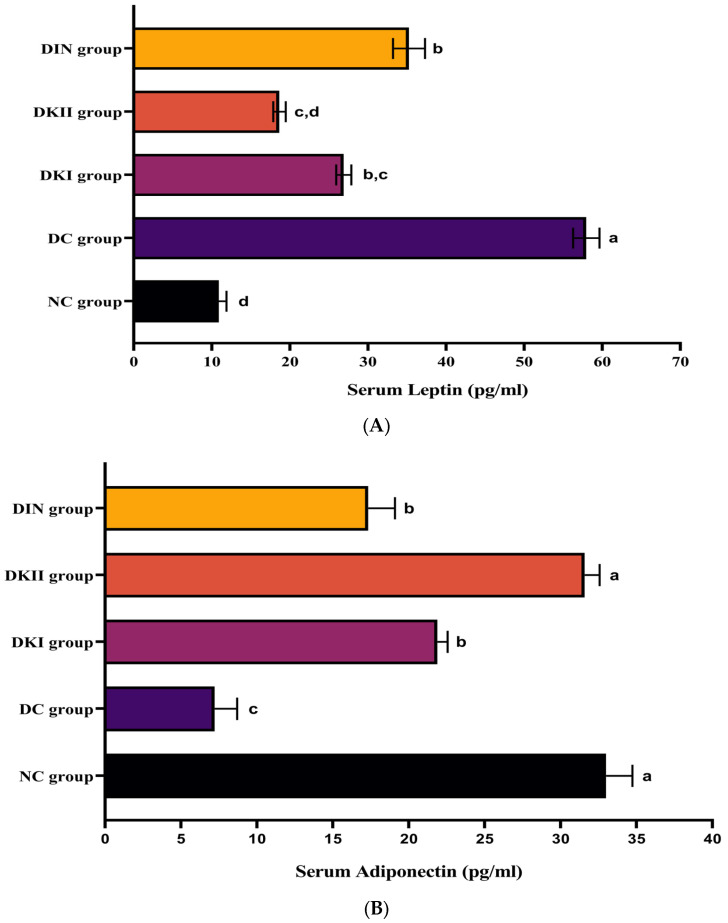
Serum values of (**A**) leptin and (**B**) adiponectin in all the studied groups. Values are expressed as mean ± SEM (*n* = 6). The means with different letters in each bar (a–d) are significantly different (*p* < 0.01), where the largest data values take the letter (a), and the smallest data values take the letter (d). NC: normal control rats, DC: diabetic control group, DKI: DC group treated with 100 mg kiwi extract, DKII: DC group treated with 200 mg/kg kiwi extract, DIN: Diabetic control group treated with insulin.

**Figure 4 ijms-24-13759-f004:**
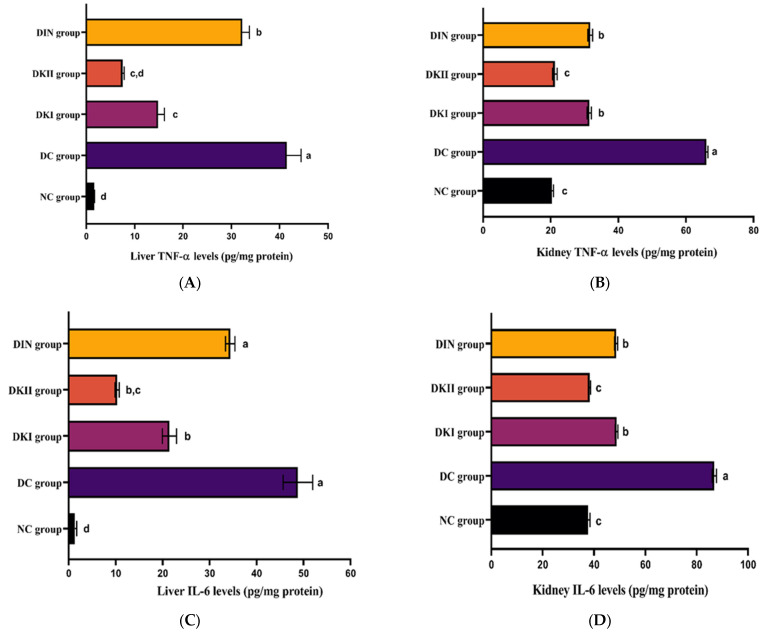
Estimation of kiwi extract effect on liver and kidney inflammatory mediators in all the studied groups. (**A**) Liver TNF-α, (**B**) kidney TNF-α, (**C**) liver INF-γ, (**D**) kidney INF-γ, (**E**) liver IL-6, (**F**) kidney IL-6, (**G**) liver NF-κB, and (**H**) kidney NF-κB. Values are expressed as mean ± SEM (*n* = 6). The means with different letters in each bar (a–d) are significantly different (*p* < 0.01), where the largest data values take the letter (a), and the smallest data values take the letter (d). NC: normal control rats, DC: diabetic control group, DKI: DC group treated with 100 mg kiwi extract, DKII: DC group treated with 200 mg/kg kiwi extract, DIN: DC group treated with insulin.

**Figure 5 ijms-24-13759-f005:**
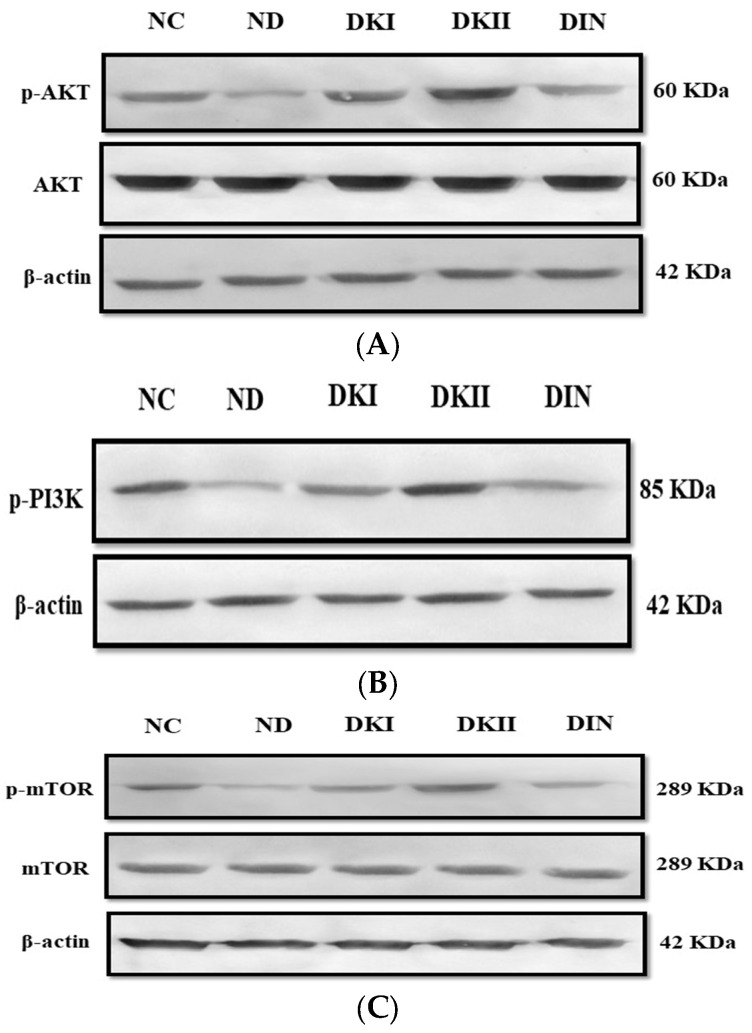
Insulin signaling pathway protein expression profile in the liver of all studied groups. (**A**) p-AKT, AKT, and β-actin; (**B**) p-PI3K and β-actin; and (**C**) p-mTOR, mTOR, and β-actin. NC: normal control rats, DC: diabetic control group, DKI: DC group treated with 100 mg kiwi extract, DKII: DC group treated with 200 mg/kg kiwi extract, DIN: DC group treated with insulin.

**Figure 6 ijms-24-13759-f006:**
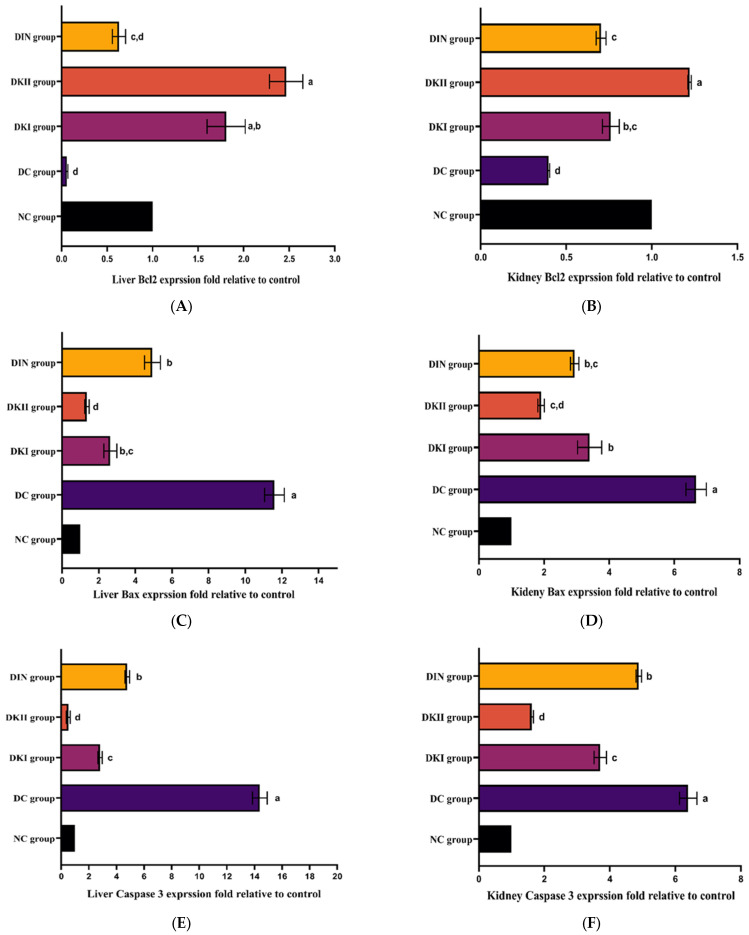
Gene expression profile of pro- and anti-apoptotic markers in the liver and kidney of all the studied groups. (**A**) Liver Bcl2, (**B**) kidney Bcl2, (**C**) liver Bax, (**D**) kidney Bax, (**E**) liver caspase 3, and (**F**) kidney caspase 3. Values are expressed as mean ± SEM (*n* = 3). The means with different letters in each bar (a–d) are significantly different (*p* < 0.01), where the largest data values take the letter (a), and the smallest data values take the letter (d). NC: normal control rats, DC: diabetic control group, DKI: DC group treated with 100 mg kiwi extract, DKII: DC group treated with 200 mg/kg kiwi extract, DIN: DC group treated with insulin.

**Figure 7 ijms-24-13759-f007:**
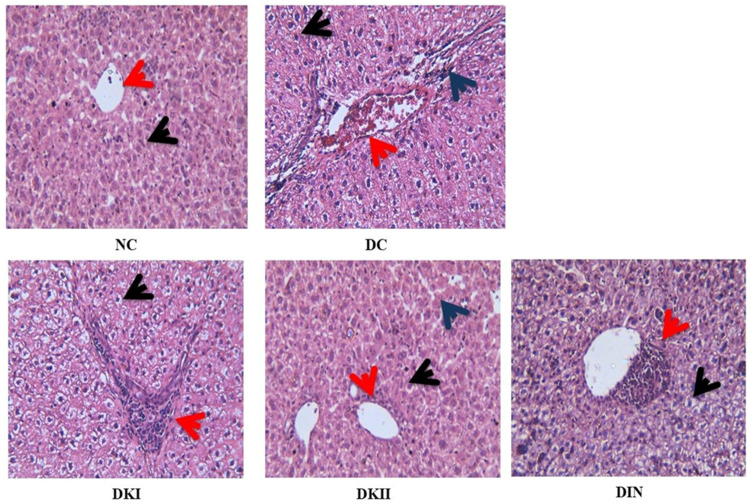
The effect of kiwi extract on the histopathology of the liver in the studied groups. The liver of rats was assessed via hematoxylin–eosin (H&E) staining at 400× magnification as the red arrows refer to mononuclear cell infiltration and the black arrows refer to fatty degeneration. NC: normal control rats, DC: diabetic control group, DKI: DC group treated with 100 mg/kg kiwi extract, DKII: DC group treated with 200 mg/kg kiwi extract, DIN: DC group treated with insulin.

**Figure 8 ijms-24-13759-f008:**
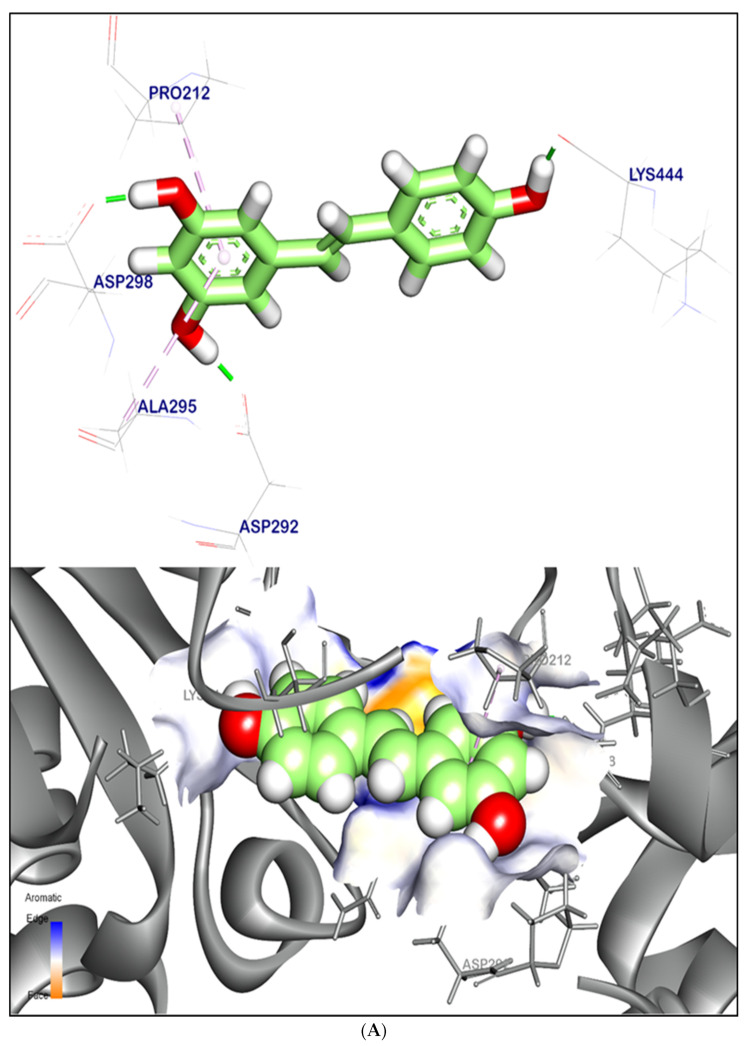
(**A**) 3D orientation and surface mapping of resveratrol against the SIRT-1 target site. (**B**) 3D orientation and surface mapping of quercetin against the SIRT-1 target site. (**C**) 3D orientation and surface mapping of chlorogenic acid against the SIRT-1 target site. (**D**) 3D orientation and surface mapping of melezitose against the SIRT-1 target site. (**E**) 2D orientation and surface mapping of quercetin, chlorogenic acid, and melezitose against the SIRT-1 target site.

**Figure 9 ijms-24-13759-f009:**
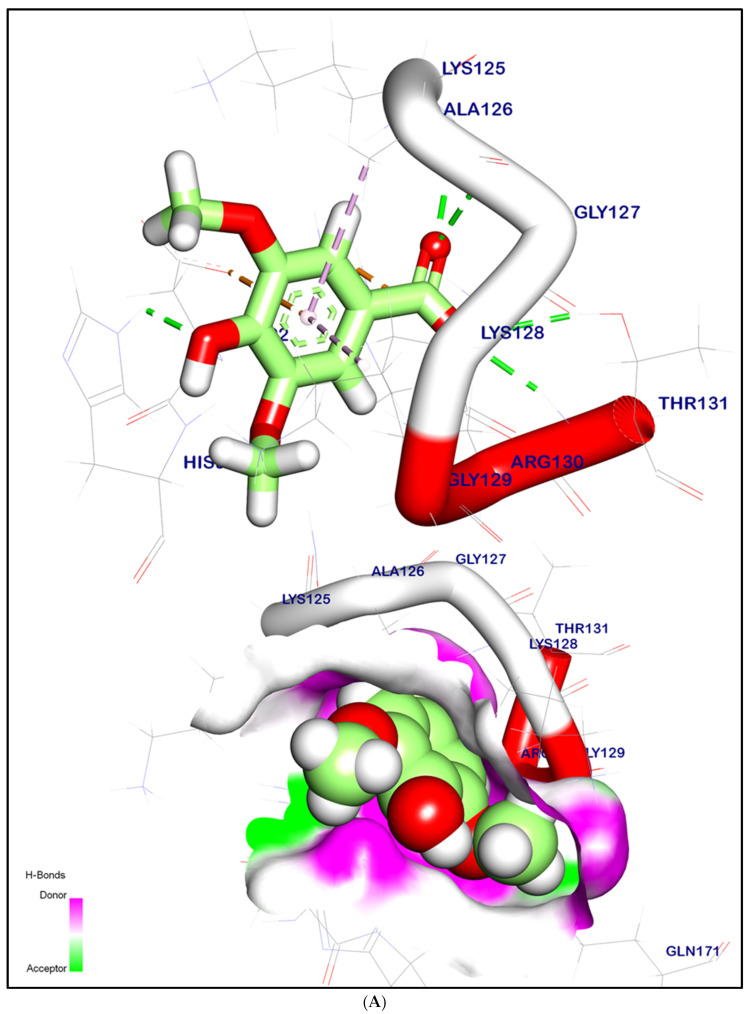
(**A**) 3D orientation and surface mapping of syringic acid against the PTEN target site. (**B**) 3D orientation and surface mapping of p-coumaric acid against the PTEN target site. (**C**) 3D orientation and surface mapping of caffeic acid against the PTEN target site. (**D**) 3D orientation and surface mapping of ferulic acid against the PTEN target site. (**E**) 2D orientation and surface mapping of syringic acid, p-coumaric acid, caffeic acid, and ferulic acid against the PTEN target site, respectively.

**Figure 10 ijms-24-13759-f010:**
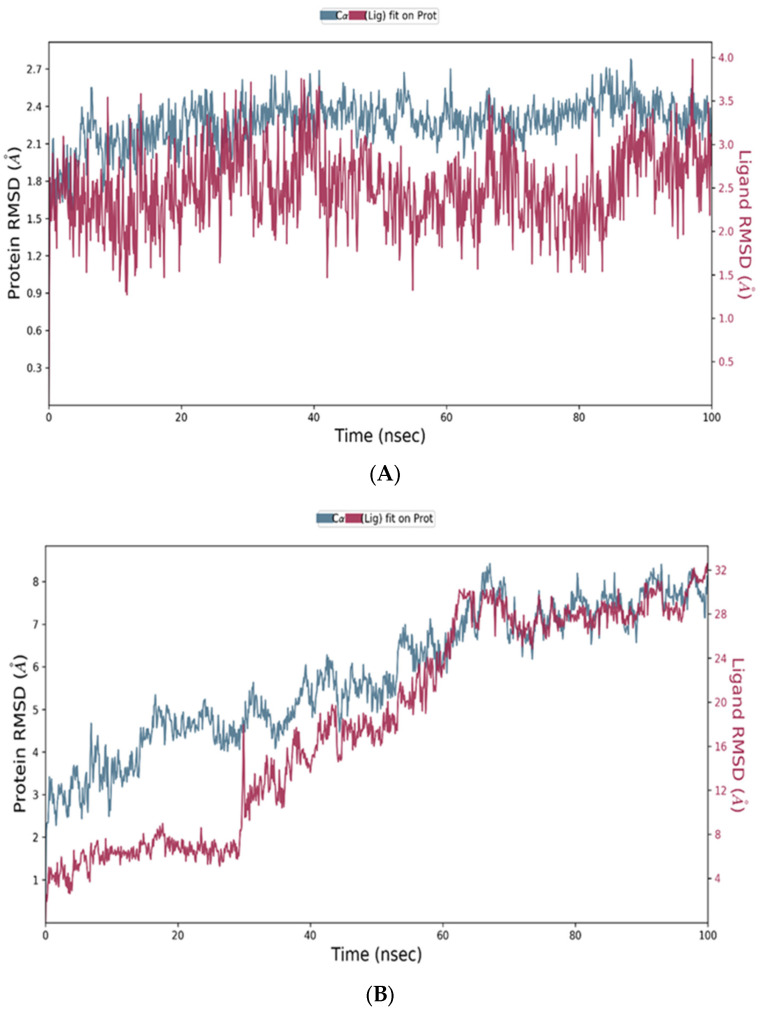
(**A**) The RMSD of the syringic acid PTEN complex for 100 ns. (**B**) The RMSD of the quercetin SIRT-1 complex for 100 ns. (**C**) The RMSF of the syringic acid PTEN complex for 100 ns. (**D**) The RMSF of the quercetin SIRT-1 complex for 100 ns. The protein residue that interact with ligand are marked with green-colored vertical bars.

**Figure 11 ijms-24-13759-f011:**
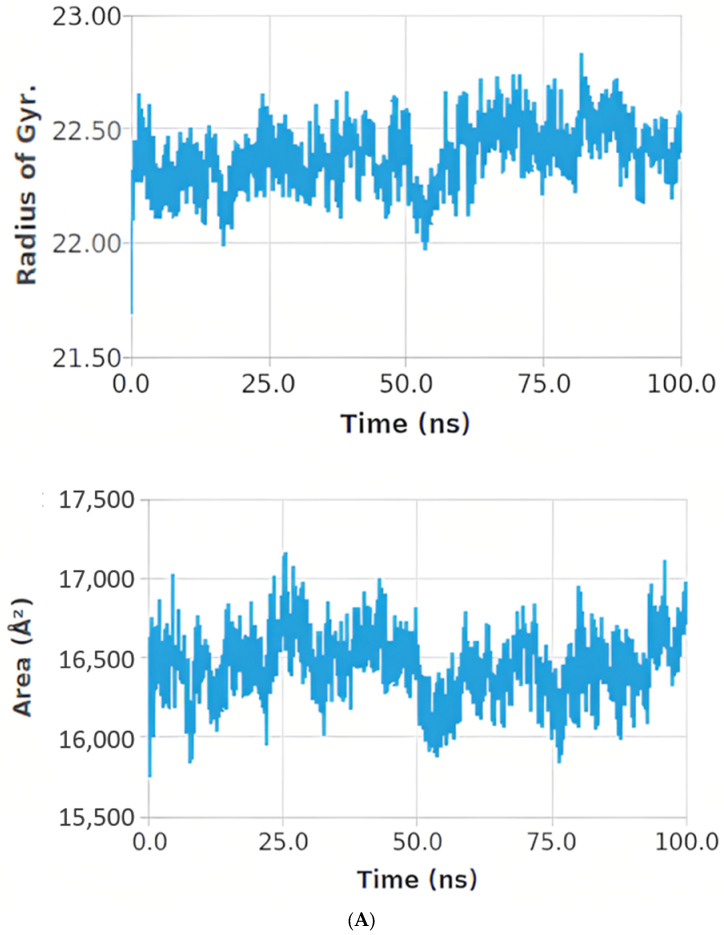
(**A**) The Rg and SASA of the syringic acid PTEN complex for 100 ns. (**B**) The Rg and SASA of the quercetin SIRT-1 complex for 100 ns.

**Figure 12 ijms-24-13759-f012:**
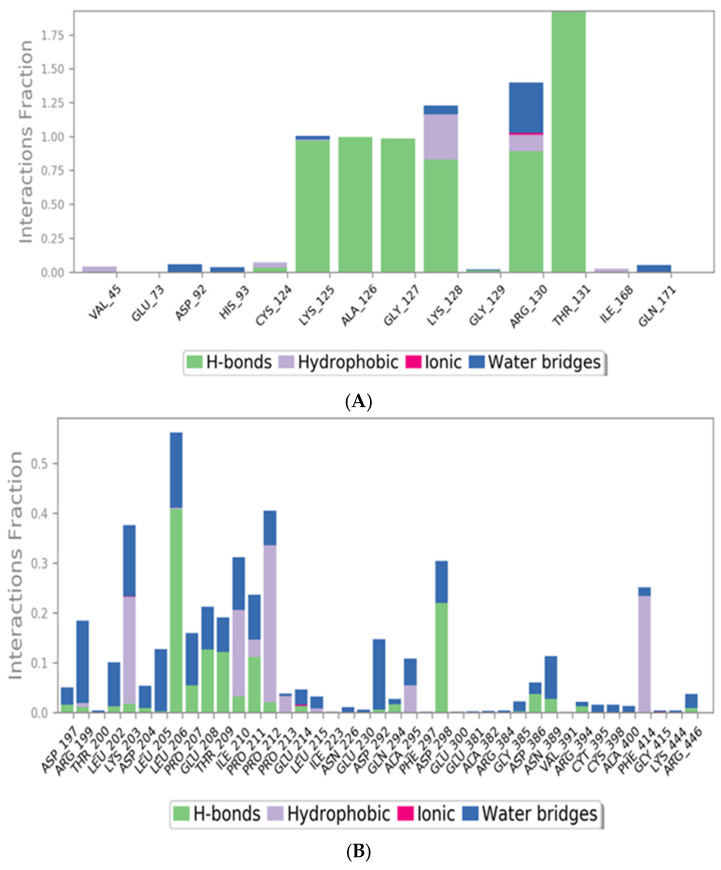
(**A**) Histogram describing the binding interactions of the syringic acid PTEN complex during the simulation time (100 ns). (**B**) Histogram describing the binding interactions of the quercetin SIRT-1 complex during the simulation time (100 ns).

**Figure 13 ijms-24-13759-f013:**
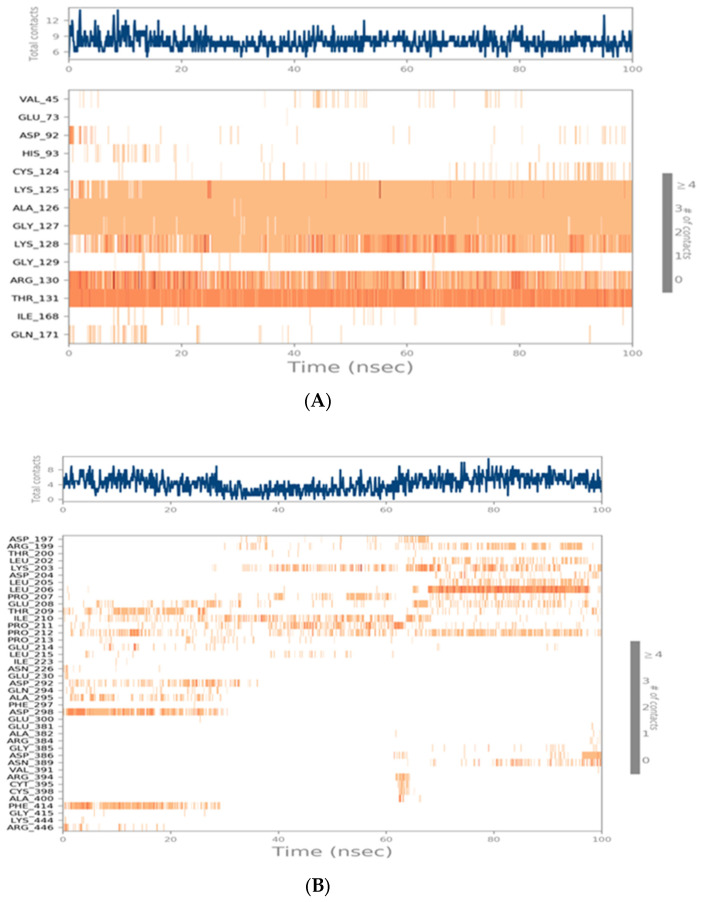
(**A**) Heat map describing the total number of interactions within the syringic acid PTEN complex during the 100 ns. (**B**) Heat map describing the total number of interactions within the quercetin SIRT-1 complex during the 100 ns. A timeline representation of the interactions and contacts (H-bonds, hydrophobic, ionic, water bridge). The top panel shows the total number of specific contacts the protein makes the ligand over the course of trajectory. The bottom panel shows which residue interact with the ligand in each trajectory frame. Some residues make more than one specific contact with the ligand, which is represented by a darker shade of orange, according to the scale to the right of the plot.

**Figure 14 ijms-24-13759-f014:**
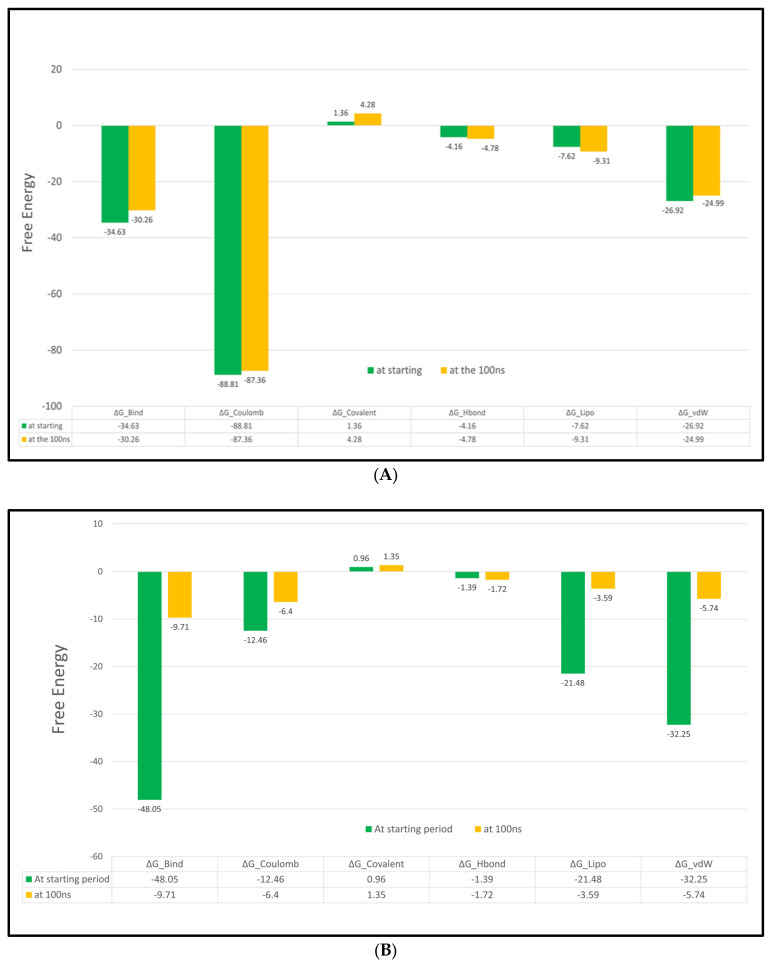
(**A**) MM-GBSA energies for the syringic acid PTEN complex (kcal/mol). Lipo: lipophilic energy; vdW: van der Waals energy. (**B**) MM-GBSA energies for the quercetin SIRT-1 complex (kcal/mol). Lipo: lipophilic energy; vdW: van der Waals nergy.

**Figure 15 ijms-24-13759-f015:**
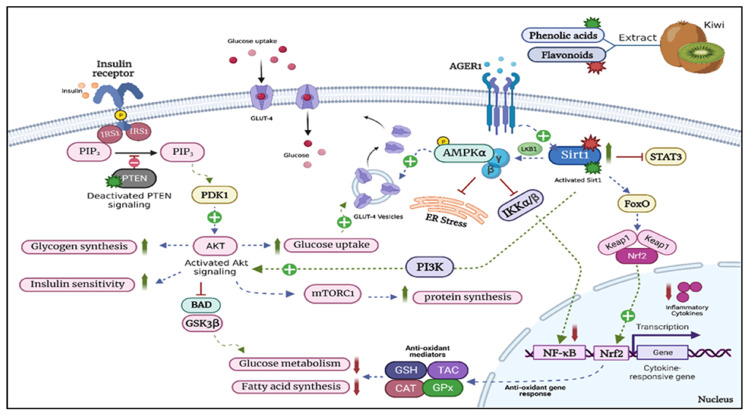
Schematic illustration for the effects of Actinidia deliciosa (kiwi) extract on pathways related to T2DM. HFD/STZ-induced hyperglycemia, dyslipidemia, cellular stress, and insulin resistance caused the inhibition of PTEN and activation of SIRT-1. (↑&⊕): upregulated targets; (↓&┴) downregulated targets. T2D Type: II diabetes, HFD: high-fat diet, STZ: streptozotocin; PTEN: phosphatase and tensin homolog, IRS-1: insulin receptor substrate 1, PIP2: phosphatidylinositol 4,5-bisphosphate, PIP3: phosphatidylinositol (3,4,5)-trisphosphate, PDK1: pyruvate dehydrogenase kinase 1, AKT: serine/threonine-protein kinase, BAD: BCL2-associated agonist of cell death, GSK3β: glycogen synthase kinase-3 beta, AGER1: advanced glycated end-product receptor 1, GLUT4: glucose transporter type 4, SIRT1: silent information regulator 1, STAT3: signal transducer and activator of transcription 3, FOXO: forkhead box O transcription factors; Keap1: Kelch-like ECH-associated protein 1, Nrf2: nuclear factor erythroid 2-related factor 2, NF-κβ: nuclear factor kappa-beta, PI3K: phosphoinositide 3-kinases, mTORC1: mammalian target of rapamycin complex 1, LKB1: liver kinase B1 also known as serine/threonine kinase 11 (STK11), AMPK: AMP-activated protein kinase, IKKα/β: IKK kinases, ER: endoplasmic reticulum, GSH: glutathione, GPx: glutathione peroxidase, CAT: catalase, and TAC: total antioxidant capacity.

**Table 1 ijms-24-13759-t001:** Evaluation of the kiwi extract increase on blood glucose and insulin levels.

Animal Groups	Glucose (mg/dL)	Insulin (µIU/mL)	HbA1c (%)	HOMA-IR
NC group	92.74 ± 3.49 ^d^	5.16 ± 0.26 ^c^	5.54 ± 0.11 ^d^	1.18 ± 0.68 ^d^
DC group	371.86 ± 2.42 ^a^	78.23 ± 2.23 ^a^	11.69 ± 0.42 ^a^	71.8 ± 2.74 ^a^
DKI group	115.70 ± 2.32 ^b^	14.65 ± 2.27 ^b^	6.65 ± 0.36 ^b,c^	4.2 ± 0.86 ^c^
DKII group	96.59 ± 3.46 ^d^	11.13 ± 1.34 ^b^	5.80 ± 0.76 ^d,c^	2.7 ± 0.81 ^c,d^
DIN group	132.57 ± 3.38 ^b^	73.12 ± 3.25 ^a^	9.54 ± 0.38 ^b^	23.9 ± 0.91 ^b^

Values are expressed as mean ± SEM (*n* = 6). The means with different letters in each bar (a–d) are significantly different (*p* < 0.01), where the largest data values take the letter (a), and the smallest data values take the letter (d). NC: normal control rats, DC: diabetic control group, C group, DKI: DC group treated with 100 mg kiwi extract, DKII: DC group treated with 200 mg/kg kiwi extract, DIN: DC group treated with insulin.

**Table 2 ijms-24-13759-t002:** Levels of LPO and GSH in renal and hepatic tissues of the experimental animal groups.

Animal Groups	LPO (nmol/mg Protein)		GSH (ng/mg Protein)	
	Kidney	Liver	Kidney	Liver
NC group	51.25 ± 1.01 ^d^	5.71 ± 0.55 ^d^	5.11 ± 0.07 ^a^	50.98 ± 2.73 ^a^
DC group	102.69 ± 1.45 ^a^	39.13 ± 0.65 ^a^	1.49 ± 0.06 ^d^	9.22 ± 0.60 ^d^
DKI group	58.92 ± 2.68 ^b,c^	8.27 ± 1.75 ^b,c^	4.23 ± 1.97 ^b,c^	43.86 ± 1.59 ^b,c^
DKII group	52.35 ± 2.38 ^c,d^	6.17 ± 0.75 ^c,d^	5.05 ± 1.07 ^a,b^	47.32 ± 2.44 ^a^
DIN group	60.39 ± 2.69 ^b^	8.74 ± 0.61 ^b^	3.96 ± 0.08 ^c^	40.48 ± 2.52 ^c^

Values are expressed as mean ± SEM (*n* = 6). The means with different letters in each bar (a–d) are significantly different (*p* < 0.01), where the largest data values take the letter (a), and the smallest data values take the letter (d). NC: normal control rats, DC: diabetic control group, DKI: DC group treated with 100 mg kiwi extract, DKII: DC group treated with 200 mg/kg kiwi extract, DIN: DC group treated with insulin.

**Table 3 ijms-24-13759-t003:** Activities of SOD, CAT, GPx, GST, and GR in the renal tissues of the experimental animal groups.

Animal Groups	SOD (Units/mg Protein)	CAT (n mol of H_2_O_2_ Released/min/mg Protein)	GPx (µ moles of GSH Oxidized/min/mg Protein)	GST (µ moles of CDNB Utilized/min/mg Protein)	GR (μg GSSG Utilized/min/mg Protein)
NC group	8.05 ± 0.17 ^a^	6.12 ± 0.15 ^a^	30.32 ± 0.83 ^a^	121.73 ± 1.52 ^a^	69.91 ± 0.47 ^a^
DC group	1.84 ± 0.09 ^d^	2.12 ± 0.03 ^d^	15.08 ± 0.17 ^d^	56.60 ± 0.90 ^d^	19.10 ± 0.47 ^d^
DKI group	6.97 ± 1.28 ^b,c^	5.67 ± 0.94 ^b,c^	25.28 ± 1.62 ^b,c^	114.24 ± 2.91 ^b,c^	63.75 ± 2.63 ^b,c^
DKII group	7.81 ± 1.25 ^a,b^	6.00 ± 0.93 ^a,b^	29.43 ± 2.71 ^a,b^	118.82 ± 1.92 ^a,b^	68.31 ± 1.85 ^a,b^
DIN group	6.36 ± 1.01 ^c^	4.91 ± 1.03 ^c^	23.57 ± 1.42 ^c^	108.16 ± 2.73 ^c^	59.51 ± 1.87 ^c^

Values are expressed as mean ± SEM (*n* = 6). The means with different letters in each bar (a–d) are significantly different (*p* < 0.01), where the largest data values take the letter (a), and the smallest data values take the letter (d). NC: normal control rats, DC: diabetic control group, DKI: DC group treated with 100 mg kiwi extract, DKII: DC group treated with 200 mg/kg kiwi extract, DIN: DC group treated with insulin.

**Table 4 ijms-24-13759-t004:** Activities of SOD, CAT, GSH-Px, GST, and GR in the hepatic tissues of the experimental animal groups.

Animal Groups	SOD (Units/mg Protein)	CAT (n mol of H_2_O_2_ Released/min/mg Protein)	GPx (µ moles of GSH Oxidized/min/mg Protein)	GST (µmoles of CDNB Utilized/min/mg Protein)	GR (μg GSSG Utilized/min/mg Protein)
NC group	18.35 ± 0.82 ^a^	179.59 ± 5.82 ^a^	25.04 ± 1.05 ^a^	2.02 ± 0.14 ^a^	3.82 ± 0.09 ^a^
DC group	5.76 ± 0.59 ^d^	84.42 ± 2.36 ^c^	11.68 ± 0.53 ^d^	0.53 ± 0.06 ^d^	1.88 ± 0.04 ^c^
DKI group	13.89 ± 1.84 ^b,c^	163.49 ± 2.61 ^b^	21.81 ± 1.28 ^b,c^	1.72 ± 1.09 ^b^	3.41 ± 0.43 ^b^
DKII group	16.23 ± 1.92 ^a,b^	175.98 ± 3.80 ^a^	24.96 ± 1.01 ^a,b^	1.96 ± 0.97 ^a^	3.80 ± 0.08 ^a^
DIN group	11.75 ± 1.79 ^c^	160.10 ± 1.98 ^b^	20.81 ± 1.29 ^c^	1.51 ± 0.67 ^c^	3.38 ± 0.29 ^b^

Values are expressed as mean ± SEM (*n* = 6). The means with different letters in each bar (a–d) are significantly different (*p* < 0.01), where the largest data values take the letter (a), and the smallest data values take the letter (d). NC: normal control rats, DC: diabetic control group, DKI: DC group treated with 100 mg kiwi extract, DKII: DC group treated with 200 mg/kg kiwi extract, DIN: DC group treated with insulin.

**Table 5 ijms-24-13759-t005:** Evaluation of kiwi extract upshot on serum lipid profile.

Animal Groups	Total Cholesterol (mg/dL)	HDL-Cholesterol (mg/dL)	LDL-Cholesterol (mg/dL)	Triglyceride (mg/dL)	Total Lipid (mg/dL)
NC group	100.95 ± 2.88 ^d^	50.87 ± 1.69 ^a^	32.25 ± 1.73 ^d^	63.56 ± 2.34 ^d^	399.44 ± 3.12 ^d^
DC group	236.20 ± 3.83 ^a^	30.51 ± 1.90 ^d^	69.12 ± 2.20 ^a^	255.08 ± 2.74 ^a^	898.66 ± 2.28 ^a^
DKI group	116.46 ± 5.64 ^c^	44.78 ± 1.66 ^b,c^	44.86 ± 1.29 ^b,c^	81.59 ± 4.17 ^c^	436.07 ± 3.23 ^c^
DKII group	103.30 ± 4.35 ^c,d^	49.13 ± 2.23 ^a,b^	33.46 ± 1.98 ^d^	67.53 ± 5.47 ^c,d^	400.51 ± 3.85 ^d^
DIN group	137.20 ± 2.45 ^b^	42.59 ± 2.14 ^c^	46.77 ± 1.75 ^b^	143.09 ± 3.06 ^b^	439.02 ± 2.62 ^b^

Values are expressed as mean ± SEM (*n* = 6). The means with different letters in each bar (a–d) are significantly different (*p* < 0.01), where the largest data value take the letter (a), and the smallest data values take the letter (d). NC: normal control rats, DC: diabetic control group, DKI: DC group treated with 100 mg kiwi extract, DKII: DC group treated with 200 mg/kg kiwi extract, DIN: DC group treated with insulin.

**Table 6 ijms-24-13759-t006:** Evaluation of kiwi extract upshot on serum renal and hepatic functions.

Animal Groups	Creatinine (mg/dL)	Urea (mg/dL)	Uric Acid (mg/dL)	ALT (U/mL)	AST (U/mL)	ALP (IU/L)
NC group	0.75 ± 0.14 ^d^	24.30 ± 1.99 ^d^	1.49 ± 0.06 ^c^	17.52 ± 1.27 ^d^	37.29 ± 1.68 ^c^	44.89 ± 1.86 ^c^
DC group	7.19 ± 0.54 ^a^	71.20 ± 1.9 ^a^	8.87 ± 0.70 ^a^	71.36 ± 1.44 ^a^	117.87 ± 1.66 ^a^	123.52 ± 1.65 ^a^
DKI group	1.02 ± 0.35 ^c^	29.09 ± 2.72 ^c^	2.36 ± 0.84 ^b^	21.02 ± 2.29 ^b,c^	55.91 ± 1.44 ^a^	53.26 ± 1.93 ^b^
DKII group	0.77 ± 0.43 ^d^	24.25 ± 2.84 ^c,d^	1.51 ± 0.13 ^c^	18.18 ± 1.24 ^c,d^	38.53 ± 1.04 ^c^	46.10 ± 1.67 ^c^
DIN group	1.25 ± 0.67 ^b^	34.16 ± 2.09 ^b^	2.59 ± 0.39 ^b^	25.89 ± 1.48 ^b^	57.63 ± 1.62 ^b^	55.19 ± 1.64 ^b^

Values are expressed as mean ± SEM (*n* = 6). The means with different letters in each bar (a–d) are significantly different (*p* < 0.01), where the largest data values take the letter (a), and the smallest data values take the letter (d). NC: normal control rats, DC: diabetic control group, DKI: DC group treated with 100 mg kiwi extract, DKII: DC group treated with 200 mg/kg kiwi extract, DIN: DC group treated with insulin.

**Table 7 ijms-24-13759-t007:** DG and RMSD interactions of the tested metabolites against the SIRT-1 and PTEN target sites.

Targets	Tested Compounds	RMSD Value (Å)	Docking (Affinity) Score (kcal/mol)	Interactions	
				H.B Pi Interaction	
SIRT-1	Resveratrol	0.75	−6.90	3	2
	Quercetin	1.03	−6.50	3	6
	Chlorogenic acid	1.73	−6.04	4	3
	Melezitose	1.69	−7.65	4	0
PTEN	Syringic acid	1.54	−6.07	4	3
	p-Coumaric acid	1.76	−5.98	4	2
	Caffeic acid	1.46	−6.06	4	2
	Ferulic acid	1.53	−6.15	4	3

## Data Availability

The manuscript contains the datasets obtained during and/or analyzed during the current research.

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
