# Peer review of "Actinidia deliciosa Extract as a Promising Supplemental Agent for Hepatic and Renal Complication-Associated Type 2 Diabetes (In Vivo and In Silico-Based Studies)"

_ijms, 2023, doi:10.3390/ijms241813759_

Round 1
Reviewer 1 Report
The manuscript assessed the effects of kiwi fruit extract on adipocytokines, antioxidant enzymes, inflammatory parameters, as well as anti-apoptotic gene expression biomarkers along with protein levels of mTOR, Akt, and PI3K in the tissues (serum or kidney or liver ) of type 2 diabetic rats. And molecular docking and dynamics simulation studies revealed that quercetin, chlorogenic acid, and melezitose as components of kiwi extract docked well with SIRT-1. Phenolic acids of kiwi extract especially syringic acid, P-coumaric acid, caffeic acid, and ferulic acid have great ability to inhibit PTEN active site.
1. The abbreviations should be described in detail when they show at the first time.
2. The units should be standardized following the writer’s guidance.
3. What’s the amount of kiwifruit you used to make the 100mg/kg, 200mg/kg kiwi extract, please clarify the process in detail in the method.
4. Have you ever checked the components of the kiwi ethanol extract in your hand? What’s the main components, how much specific components are there at the treatment doses 100mg/kg, 200mg/kg? Especially for those compounds you analyzed in molecular docking.
5. In the method, the information of all related kits should be provided, presented in consistent format and the method should be also clarified briefly at least.
6. What’s the relationship of SIRT-1 and PTEN with type 2 diabetes? In which organ? Why do you choose them to do docking, please clarify. The docking section is lack of connection with the previous part.
7. Why liver and kidney rather than pancreas or other organs were chosen for apoptotic, inflammatory analysis?
8. Please make it clear in the figures or figure legends that which tissue you tested for the respective experiment.
9. In figure 1, the statistical significance should be analyzed, the legend in figure should be described.
10. All the comparisons in figures and tables are suggested to be relabeled in another way, for example *:p<0.05 **: p<0.01 *** p<0.001 so that people know how significant they are. And what’s the control for the comparison, the NC group or DC group.
11. The histology changes seem not obvious as described.
12. It is very striking that the kiwi fruit extract has stronger effect than insulin, what’s the possible reason and mechanisms behind? Will the specific main components in kiwi affect the phosphorylation levels of PI3K, mTOR? Will they affect the antioxidant, inflammatory status? how much of these signaling pathway contribute to the diabetes?
13. The manuscript will benefit from more logic connection by improving the abstract and results.

The expression should be improved.
Author Response
We are grateful to you for the insightful comments on our paper. We have been able to incorporate changes to reflect most of the suggestions provided. We want to inform you that all edits and amends required, have been met and achieved and we genuinely thank you for it as it formed a great added value to the work. We have highlighted the changes within the manuscript. Here is a point-by-point response to the reviewers' comments and concerns.
Point 1. The abbreviations should be described in detail when they show at the first time.
Response:
We sincerely appreciate you doing so because it vastly improved the work. According to your comment, we have revised my manuscript, and all the revised parts are marked with bright yellow.
Point 2. The units should be standardized following the writer’s guidance.
Response:
Thank you for your notes, all units used are changed to the standard units.
Point 3. What’s the amount of kiwifruit you used to make the 100 mg/kg, 200 mg/kg kiwi extract, please clarify the process in detail in the method.
Response:
Thank you for your recommendation, we added this part to method section.
The administered doses of kiwi extract were 100 mg/kg/day and 200 mg/kg/day. The administration of doses occurred consistently at a fixed time each day. The kiwi extract solutions were made by dissolving the appropriate dosage of lyophilized powder, equivalent to either 100 mg or 200 mg of extract, in distilled water immediately before use. The amount of extract used for each rat was determined based on their body weight. Subsequently, the rats consumed the prepared solutions via oral gavage (the revised parts are marked with bright yellow in the manuscript).
In more detail, in case of stock extract solution, which is sparingly soluble in water that intended to be administered at a dose of 100 mg/kg, the concentration of solution could be used as follows. For example, if the dose of rats is 100 mg/kg, a stock solution of 100 mg/ml has to be prepared. If the dose is 200 mg/kg, prepare a stock solution of 200 mg/ml and so on. From the stock solution of 100 mg/ml, an appropriate volume of the extract solution could be administered according to the body weight of animals. For example, if the body weights of rats are 300 g, it is suggested to ingest 0.30 ml from the stock solution (100 mg/ml).
Point 4. Have you ever checked the components of the kiwi ethanol extract in your hand? What’s the main components, how much specific components are there at the treatment doses 100mg/kg, 200mg/kg? Especially for those compounds you analyzed in molecular docking.
Response:
The components of the kiwi ethanol extract was checked in previous research with Dr. Heba Sayed Mostafa (Food Science Department, Faculty of Agriculture, Cairo University, Giza, Egypt). After preparation of the ethanoic extract for the kiwi extract fruit's peel and flesh, HPLC was utilized to identify the phenolic compounds in the extract. In addition to the estimation of the total phenolic and flavonoid contents.
References
- El Azab, E. F.; Mostafa, H. S. Phytochemical analysis and antioxidant defense of kiwifruit (Actinidia deliciosa) against pancreatic cancer and AAPH-induced RBCs hemolysis. Food Sci. Technol. 2021, 42, e06021. https://doi.org/10.1590/fst.06021
Point 5. In the method, the information of all related kits should be provided, presented in consistent format and the method should be also clarified briefly at least.
Response:
We agree with the reviewer and all the information added, as well as the methods clarified in the method section.
Point 6. What’s the relationship of SIRT-1 and PTEN with type 2 diabetes? In which organ? Why do you choose them to do docking, please clarify. The docking section is lack of connection with the previous part.
Response:
Restricting calorie intake (calorie restriction or CR) has the potential to slow down the aging process and postpone the onset of various age-related conditions, such as diabetes. Consequently, compounds that mimic the metabolic effects of CR could serve as innovative targets for treating Type 2 Diabetes (T2DM). Sirtuin 1 (SIRT1), an enzyme that depends on the molecule NAD+ and removes acetyl groups from histones, is stimulated by CR and is closely linked to extending lifespan through CR [1].
SIRT1 plays a significant role in managing the metabolism of glucose and lipids by deacetylating numerous substrates. In pancreatic β-cells, SIRT1 has a favorable impact on insulin secretion and safeguards cells from harm caused by oxidative stress and inflammation. Moreover, SIRT1 positively influences metabolic pathways by adjusting insulin signaling. SIRT1 also regulates the release of adiponectin, manages inflammation, control glucose production, alleviates oxidative stress, and maintains proper mitochondrial function
Several substances that activate SIRT1, including resveratrol, have been shown to offer advantageous effects on maintaining stable glucose levels and enhancing the body's sensitivity to insulin in animal models where insulin resistance is present.
The SIRT1/AKT/Insulin signaling pathway plays a significant role in regulating crucial aspects of cellular function, especially concerning metabolism, cellular expansion, and the body's responsiveness to insulin. This is achieved by activating AKT and AMPKα [2], which are key players in facilitating glucose uptake through GLUT-4. The intricate interplay within this pathway is critical for ensuring appropriate reactions to insulin within the body and effectively maintaining the balance of glucose. Let's explore the significance of this pathway and its link to Type 2 Diabetes Mellitus (DM-2):
Role of SIRT1/AKT/Insulin Signaling Pathway:
- Insulin Sensitivity: The pathway enables cells to respond efficiently to insulin. Insulin binding to its receptor triggers a cascade of events that ultimately leads to the activation of AKT. Active AKT promotes the translocation of glucose transporters (such as GLUT4) to the cell membrane, facilitating glucose uptake and reducing blood glucose levels [3].
- Cell Growth and Survival: AKT, when activated, plays a central role in promoting cell growth, proliferation, and survival. It phosphorylates and inhibits factors that promote cell cycle arrest and apoptosis, contributing to cell survival and growth.
- Metabolic Regulation: AKT activation has a direct impact on metabolic processes. It phosphorylates and modulates the activity of enzymes involved in glycogen synthesis, protein synthesis, and lipid metabolism. This ensures proper utilization and storage of nutrients.
- Energy Homeostasis: SIRT1, a NAD+-dependent protein deacetylase, interacts with components of this pathway. It acts as a metabolic sensor, responding to changes in cellular energy levels. SIRT1 regulates gene expression and protein activity, impacting metabolism and energy balance.
Relation of SIRT1 with DM-2:
Type 2 Diabetes Mellitus (DM-2) is characterized by insulin resistance, where cells become less responsive to insulin's signaling, leading to elevated blood glucose levels. SIRT1 has emerged as a critical player in the development of insulin resistance and DM-2:
- Insulin Sensitivity: SIRT1 levels are reduced in insulin-resistant states. This reduction in SIRT1 activity can impair insulin signaling, leading to decreased glucose uptake and insulin resistance.
- Inflammation and Oxidative Stress: SIRT1 possesses anti-inflammatory and antioxidant properties. Reduced SIRT1 activity can lead to chronic inflammation and increased oxidative stress, both of which contribute to insulin resistance and DM-2.
- Mitochondrial Function: SIRT1 interacts with factors involved in mitochondrial function. Impaired SIRT1 activity may disrupt mitochondrial health, leading to reduced energy production and insulin resistance.
- Regulation of Gluconeogenesis: SIRT1 normally suppresses gluconeogenic genes in the liver. Reduced SIRT1 activity can lead to excessive glucose production by the liver, contributing to hyperglycemia in DM-2.
- Role in Adipose Tissue: SIRT1 is involved in adipose tissue function and differentiation. Dysregulation of SIRT1 in adipocytes can lead to altered adipokine secretion and contribute to insulin resistance.
Overall, the SIRT1/AKT/Insulin signaling pathway and its interplay with SIRT1 are crucial in maintaining insulin sensitivity and metabolic balance. Dysregulation of this pathway, particularly reduced SIRT1 activity, can contribute to the development of insulin resistance and Type 2 Diabetes Mellitus. Understanding and targeting these mechanisms holds promise for developing therapeutic approaches for DM-2 management.
The relationship between PTEN (Phosphatase and Tensin Homolog) and Type 2 Diabetes (T2D) is complex and involves the regulation of insulin signaling, glucose metabolism, and cell growth. PTEN is a key protein that plays a critical role in controlling cell growth, survival, and metabolism through its enzymatic activity as a lipid and protein phosphatase. Its relationship with T2D revolves around its impact on insulin sensitivity and glucose homeostasis. Here's how PTEN is linked to Type 2 Diabetes:
- Insulin Signaling and Glucose Metabolism:
- PTEN negatively regulates the insulin signaling pathway. It dephosphorylates a lipid messenger called phosphatidylinositol 3, 4, 5-trisphosphate (PIP3), which is produced downstream of insulin receptor activation. This leads to decreased activation of downstream effectors, including AKT, that are crucial for glucose uptake and utilization.
- Reduced PTEN activity can enhance insulin signaling, resulting in improved insulin sensitivity and glucose uptake in peripheral tissues (such as muscle and adipose tissue). Conversely, increased PTEN activity contributes to insulin resistance, a hallmark of T2D.
- Role in Adipose Tissue:
- PTEN activity in adipose tissue influences adipogenesis (the formation of fat cells) and adipokine secretion (hormones produced by fat cells). Dysregulation of PTEN in adipose tissue can affect adipocyte function and contribute to insulin resistance by altering the secretion of factors that modulate insulin sensitivity.
- Regulation of Cell Growth and Survival:
- PTEN is known for its tumor suppressor role due to its ability to inhibit cell growth and proliferation. However, excessive PTEN activity can lead to cell death (apoptosis). In the context of T2D, abnormal cell death can impact pancreatic beta cells, which produce and secrete insulin. Loss of beta cells contributes to impaired insulin secretion and the worsening of diabetes.
- Interaction with mTOR Pathway:
- The PTEN/AKT pathway interacts with the mTOR (mammalian target of rapamycin) pathway, which is crucial for cellular growth and metabolism. Dysregulation of this interaction can influence insulin sensitivity and contribute to insulin resistance.
- Obesity and Inflammation:
- PTEN has been implicated in the regulation of adipose tissue inflammation and obesity-associated insulin resistance. Obesity, a major risk factor for T2D, is characterized by chronic low-grade inflammation, and PTEN's role in this context may impact insulin sensitivity.
In summary, PTEN's involvement in regulating insulin signaling, glucose metabolism, and cell growth has significant implications for the development and progression of Type 2 Diabetes. Dysregulation of PTEN activity can lead to impaired insulin sensitivity, and disrupted glucose homeostasis, and contribute to the underlying mechanisms of insulin resistance commonly observed in T2D. Understanding the intricate interactions between PTEN and diabetes-related pathways may provide insights into potential therapeutic approaches for managing Type 2 Diabetes.
So, activation of the SIRT1/AKT/Insulin signaling pathway and the inhibition of PTEN hold promise in the context of managing Type 2 Diabetes (T2D). Let's discuss the roles of pathway activators and PTEN inhibitors in relation to T2D:
SIRT1/AKT/Insulin Signaling Pathway Activators:
- Improved Insulin Sensitivity: Activators of this pathway, such as resveratrol, can enhance insulin sensitivity in target tissues like muscle and adipose tissue. This means cells become more responsive to insulin, resulting in better glucose uptake and utilization [1].
- Glucose Homeostasis: Activation of AKT through this pathway promotes the translocation of glucose transporters (e.g., GLUT4) to cell membranes, facilitating glucose uptake. This helps maintain stable blood glucose levels.
- Cell Survival and Function: SIRT1 activators protect pancreatic β-cells from oxidative stress and inflammation. This preserves their function and insulin secretion ability, crucial for managing T2D.
- Metabolic Regulation: Activated SIRT1 can influence various metabolic processes, including gluconeogenesis (glucose production) in the liver, thereby contributing to glucose control.
PTEN Inhibitors:
- Enhanced Insulin Signaling: Inhibiting PTEN can amplify insulin signaling by increasing levels of phosphatidylinositol 3, 4, 5-trisphosphate (PIP3), a crucial activator of AKT. This promotes better glucose uptake and utilization by cells[4].
- Insulin Sensitivity: PTEN inhibitors can improve insulin sensitivity, helping combat the insulin resistance seen in T2D [5].
- Beta-Cell Function: Inhibition of PTEN may help protect pancreatic β-cells from apoptosis (cell death) and improve insulin secretion, thus supporting glucose regulation.
- Cell Growth and Survival: By inhibiting PTEN, the mTOR pathway can be modulated, which impacts cell growth and survival. This could have implications for managing insulin resistance and T2D progression [6].
- Adipose Tissue Function: PTEN inhibition might contribute to healthier adipose tissue function and reduce inflammation, further aiding insulin sensitivity.
It's important to note that while these approaches hold potential, the development of therapies targeting the SIRT1/AKT/Insulin pathway and PTEN requires thorough research and consideration of potential side effects. Additionally, individual responses to these interventions may vary, and comprehensive clinical studies are needed to assess their effectiveness and safety in managing Type 2 Diabetes [7].
References
- Kitada, M.; Koya, D. SIRT1 in type 2 diabetes: mechanisms and therapeutic potential. Diabetes & metabolism journal. 2013, 37(5), 315-325. https://doi.org/10.4093/dmj.2013.37.5.315
- Cunningham, R. P.; Sheldon, R. D.; Rector, R. S. The emerging role of hepatocellular eNOS in Non-alcoholic fatty liver disease development. Frontiers in Physiology. 2020, 11, 767. https://doi.org/10.3389/fphys.2020.00767
- Li, M.; Chi, X.; Wang, Y.; Setrerrahmane, S.; Xie, W.; Xu, H. (2022). Trends in insulin resistance: Insights into mechanisms and therapeutic strategy. Signal Transduction and Targeted Therapy. 2022, 7(1), 216. https://doi.org/10.1038/s41392-022-01073-0
- Butler, M.; McKay, R. A.; Popoff, I. J.; Gaarde, W. A.; Witchell, D.; Murray, S. F.; Monia, B. P. Specific inhibition of PTEN expression reverses hyperglycemia in diabetic mice. Diabetes. 2002, 51(4), 1028-1034. https://doi.org/10.2337/diabetes.51.4.1028
- Li, Y. Z.; Di Cristofano, A.; Woo, M. Metabolic role of PTEN in insulin signaling and resistance. Cold Spring Harbor Perspectives in Medicine. 2020, 10(8). https://doi.org/10.1101%2Fcshperspect.a036137
- Carracedo, A.; Pandolfi, P. P. The PTEN–PI3K pathway: of feedbacks and cross-talks. Oncogene. 2008, 27(41), 5527-5541. https://doi.org/10.1038/onc.2008.247
- Pulido, R. PTEN inhibition in human disease therapy. Molecules. 2018, 23(2), 285. https://doi.org/10.3390/molecules23020285
Point 7. Why liver and kidney rather than pancreas or other organs were chosen for apoptotic, inflammatory analysis?
Response:
Kiwi fruit has been reported to exhibit anti-inflammatory activity; however, its protective effect against inflammation and its trigger factor, i.e., oxidative stress, and the related sequelae, i.e., apoptosis in the liver and kidney in T2DM, are unknown. So, we tried to identify its anti-obesity, anti-insulin resistance, anti-inflammatory, anti-apoptotic, and anti-oxidative stress effects in liver and kidney tissues of HFD/STZ-induced T2DM in rats. Moreover, we analyzed the expressions of key proteins of the insulin pathway, especially those that are involved in cellular proliferation, in the livers of experimental animals. And we planned to complete the antidiabetic profile of kiwi extract in all other organs in further study to figure out its mechanism of action in all organs.
In fact, diabetes increases the risk of liver-related disorders [1]. It is important to note that liver failure plays a significant role in the mortality associated with T2DM. Specifically, a prospective cohort analysis revealed that cirrhosis was responsible for 12.5% of all recorded deaths related to T2DM [2]. In a similar vein, it was found that approximately 20-30% of patients diagnosed with T2DM exhibited signs of renal impairment. Regrettably, the coexistence of diabetes and chronic kidney disease is frequently associated with a rise in both morbidity and death rates [3]. Unfortunately, the antidiabetic drugs cannot repair diabetes problems, and extended use may cause drug resistance and severe renal injury.
References
- Blachier, M.; Leleu, H.; Peck-Radosavljevic, M.; Valla, D.C.; Roudot-Thoraval, F. The burden of liver disease in Europe: A review of available epidemiological data. Hepatol.2013, 58, 593–608. https://doi.org/10.1016/j.jhep.2012.12.005
- Caldwell, S.H.; Oelsner, D.H.; Iezzoni, J.C.; Hespenheide, E.E.; Battle, E.H.; Driscoll, C.J. Cryptogenic cirrhosis: Clinical characterization and risk factors for underlying disease. Hepatology1999, 29, 664–669. https://doi.org/10.1002/hep.510290347
- Grandfils, N.; Detournay, B.; Attali, C.; Joly, D.; Simon, D.; Vergès, B.; Toussi, M.; Briand, Y.; Delaitre, O. Glucose Lowering Therapeutic Strategies for Type 2 Diabetic Patients with Chronic Kidney Disease in Primary Care Setting in France: A Cross-Sectional Study. J. Endocrinol.2013, 2013, 640632. https://doi.org/10.1155/2013/640632
Point 8. Please make it clear in the figures or figure legends that which tissue you tested for the respective experiment.
Response:
Thank you for your notes, we clarified both figures and legends.
Point 9. In figure 1, the statistical significance should be analyzed, the legend in figure should be described.
Response:
We changed figure 1 as advised.
Point 10. All the comparisons in figures and tables are suggested to be relabeled in another way, for example *:p<0.05 **: p<0.01 *** p<0.001 so that people know how significant they are. And what’s the control for the comparison, the NC group or DC group.
Response:
We appreciate your great notes. But I want to clarify this point; our study aimed to investigate the differences in the effectiveness of two doses of kiwi fruit extract and the local therapy of diabetes (insulin) in alleviating type 2 diabetes complications. So, to point out which dose is effective and safe, at the same time compare their effect with insulin, we applied a post-hoc test for multiple comparisons between groups and yielded a matrix where asterisks indicate significantly different group means at an alpha level of 0.01. and we also relate the comparison to positive (Diabetes-induced) and negative (control) groups to show how the treatment brought the values near the control values. Instead of comparing all groups with control or induction only, a post-hoc test was done to identify which treatment changed the levels to values equal to those found for the control group and which presented different values.
Point 11. The histology changes seem not obvious as described.
Response:
Thanks for this observation and sincere apologies for this mistake; we uploaded the correct image.
Point 12. It is very striking that the kiwi fruit extract has stronger effect than insulin, what’s the possible reason and mechanisms behind? Will the specific main components in kiwi affect the phosphorylation levels of PI3K, mTOR? Will they affect the antioxidant, inflammatory status? how much of these signaling pathway contribute to the diabetes?
Response:
It is very striking that the kiwi fruit extract has stronger effect than insulin, what’s the possible reason and mechanisms behind? Will the specific main components in kiwi affect the phosphorylation levels of PI3K, mTOR?
Nowadays, the use of natural substances may be indicated not only in the pre-diabetic stage and in the early stage of T2DM but also in the advanced stage of this disease. Besides, kiwifruit contains fat, carbohydrates (sugar and dietary fiber), protein (lutein and zeaxanthin), vitamins A, B, C, E, and K, minerals, flavonoid, polyphenols, inositol, and carotenoids. Being rich in vitamins and antioxidants, it seems helpful for metabolic health. These antioxidants are vital components of our body that fight disease by reducing oxidative stress or nullifying the excess toxic free radicals produced under various pathological conditions like diabetes. In addition, flavonoid and polyphenols in kiwi extract show can manage diabetes mellitus by protecting pancreatic β-cells against glucose toxicity, promoting proliferation, reducing apoptosis, and inhibiting α-glucosidases or α-amylase. In addition, these compounds exhibit antioxidant anti-inflammatory activities, modulate carbohydrate and lipid metabolism, optimize oxidative stress, reduce insulin resistance, and stimulate the pancreas to secrete insulin. They also activate insulin signaling (PI3K/AKT/mTOR) and (MAPK) pathways and inhibit digestive enzymes, regulate intestinal microbiota, improve adipose tissue metabolism, inhibit glucose absorption, and inhibit the formation of advanced glycation end products. And herein we determine the effective dose and detailed mechanisms of action of kiwi extract in vivo and in silico and compare its effect with standard drug and we exhibit the beneficial use of kiwi extract as a protective and curative agent against type 2 diabetes-related complications. And the multiple therapeutic targets of kiwi extract were more than insulin due to its phytoconstituents.
Will they affect the antioxidant, inflammatory status?
Unravelling the mechanisms of action of phytochemicals is an additional important step in phytochemical research and must be comprehensively investigated in order to identify novel activators of insulin signaling and to promote effective and safe application of botanical products into the clinic. In our study, the phytocomponents of kiwi extract were able to inhibit insulin resistance and activate insulin signaling pathway that shown by the increasing of the phosphorylation of downstream insulin signaling pathway. And these changes in the insulin signaling and binding effectiveness were confirmed in silico by docking with different indicators in the pathway.
In addition, Kiwifruit is one of the more popular fruits today. Kiwifruit is known as a good source of minerals, carotenoids, vitamin C and phenolics [1,2]. Kiwifruit has a considerable antioxidant capacity [3] and thus can reduce oxidative stress [4]. In addition, antioxidants in kiwifruit can effectively modulate immune and inflammatory reactions in the body [5-8].
References
- Park, Y. S.; Leontowicz, H.; Leontowicz, M.; Namiesnik, J.; Suhaj, M. M. C. M. O.; Cvikrová, M.; Gorinstein, S. Comparison of the contents of bioactive compounds and the level of antioxidant activity in different kiwifruit cultivars. J. Food Compos. Anal.2011; 24, 963–970. https://doi.org/10.1016/j.jfca.2010.08.010
- Lim, Y. J.; Oh, C. S.; Park, Y. D.; Eom, S. H.; Kim, D. O.; Kim, U. J.; Cho, Y. SPhysiological components of kiwifruits with in vitro antioxidant and acetylcholinesterase inhibitory activities. Food Sci. Biotechnol.2014, 23,943–949. https://doi.org/10.1007/s10068-014-0127-z
- Hwang, J. S.; Cho, C. H.; Baik, M. Y.; Park, S. K.; Heo, H. J.; Cho, Y. S.; Kim, D. O. Effects of freeze-drying on antioxidant and anticholinesterase activities in various cultivars of kiwifruit (Actinidia spp.) Food Sci. Biotechnol.2017, 26,221–228. https://doi.org/10.1007/s10068-017-0030-5
- Lee, I.; Lee, B. H.; Eom, S. H.; Oh, C. S.; Kang, H.; Cho, Y. S.; Kim, D. O. Antioxidant capacity and protective effects on neuronal PC-12 cells of domestic bred kiwifruit. Korean J. Hort. Sci. Technol.2015, 33,259–267. https://doi.org/10.7235/hort.2015.14123
- Park, Y. S.; Namiesnik, J.; Vearasilp, K.; Leontowicz, H.; Leontowicz, M.; Barasch, D.; Gorinstein, S. Bioactive compounds and the antioxidant capacity in new kiwi fruit cultivars. Food Chem. 2014,165, 354–361. https://doi.org/10.1016/j.foodchem.2014.05.114
- Fiorentino, A.; D’Abrosca, B.; Pacifico, S.; Mastellone, C.; Scognamiglio, M.; Monaco, P.Identification and assessment of antioxidant capacity of phytochemicals from kiwi fruits. Agric. Food Chem. 2009, 57, 4148–4155. https://doi.org/10.1021/jf900210z
- Shin, M. S.; Park, J. Y.; Lee, J.; Yoo, H. H.; Hahm, D. H.; Lee, S. C; Kang, K. S. Anti-inflammatory effects and corresponding mechanisms of cirsimaritin extracted from Cirsium japonicum var. maackii Maxim. Bioorg. Chem. Lett.2017, 27, 3076–3080. https://doi.org/10.1016/j.bmcl.2017.05.051
- Iwasawa, H.; Morita, E.; Ueda, H.; Yamazaki, M. Influence of kiwi fruit on immunity and its anti-oxidant effects in mice. Food Sci. Technol. Res.2010, 16,135–142. https://doi.org/10.3136/fstr.16.135
How much of these signaling pathway contribute to the diabetes?
Type 2 diabetes mellitus exhibits a robust correlation with oxidative stress and inflammation, hence aligning with an elevated incidence of secondary problems related to diabetes. Natural products possessing antioxidant capabilities, have the potential to mitigate a range of impairments resulting from oxidative stress, also have been found to have potential in mitigating diabetic issues. Based on accumulated evidence, the PI3K/AKT signaling pathway is required for normal metabolism due to its characteristics, and its imbalance leads to the development of obesity and type 2 diabetes mellitus [1]. Further, mTOR serves as a crucial regulator of cellular metabolism and growth. mTOR, a crucial regulator of cellular metabolic homeostasis, accomplishes this task by integrating many physiological signals, including energy, nutritional, and hormone cues. The development of pathological diseases and end-organ problems in individuals with type 2 diabetes mellitus can be related to the dysregulation of mTOR pathway. There is a significant body of evidence indicating that two signaling pathways regulated by mTOR, namely mTORC1-p70S6 kinase 1 (S6K1) and mTORC2-protein kinase B (AKT), have a crucial impact on insulin sensitivity. Furthermore, it has been observed that the malfunctioning of these pathways leads to the onset and progression of type 2 diabetes mellitus. Also, previous study suggested that PI3K/AKT/mTOR pathway genes may participate in the development of T2DM [2].
References
- Huang, X.; Liu, G.; Guo, J.; Su, Z. The PI3K/AKT pathway in obesity and type 2 diabetes. I J. Biol. Sci. 2018, 14, 1483-1496. https://doi.org/10.7150%2Fijbs.27173
- Yin, X.; Xu, Z.; Zhang, Z.; Li, L.; Pan, Q.; Zheng, F.; Li, H. Association of PI3K/AKT/mTOR pathway genetic variants with type 2 diabetes mellitus in Chinese. Diabetes Res. Clin. Pract. 2017, 128, 127-135. https://doi.org/10.1016/j.diabres.2017.04.002
Point 13.The manuscript will benefit from more logic connection by improving the abstract and results.
Response:
Thanks a lot for this recommendation. The abstract and results were improved as advised.

Reviewer 2 Report
The scientific paper "Alleviation of Type 2 Diabetes-Related Multiple Complications in High-Fat Diet/Streptozotocin-Induced Male Rats: Insights into The in vivo and in silico Mechanisms of Actinidia deliciosa and Its Major Phyto-constituents" submitted for review is very interesting.
The results obtained by the authors are very interesting and give hope for progress in the development of treatments for type 2 diabetes.
The work is original and written accurately.
However, I think the title is a bit confusing. Please think about whether it could be rewritten to reflect the research conducted but also be easier for the reader to read. This would certainly make it more easily and widely available to both doctors and researchers.
I don't understand why the Material and Methods section is included only after the discussion of the results. I do not understand what was the intention of such a concept. I leave it to the decision of the editors to accept such a layout of the work.
On a positive note, the very good documentation and illustrations of the obtained research results deserve to be emphasized.
Author Response
First of all, we really appreciate your diligent revision.
Point: However, I think the title is a bit confusing. Please think about whether it could be rewritten to reflect the research conducted but also be easier for the reader to read. This would certainly make it more easily and widely available to both doctors and researchers.
Response:
Thank you for your suggestion. We simplified and rewrote the title as advised.
I don't understand why the Material and Methods section is included only after the discussion of the results. I do not understand what the intention of such a concept was. I leave it to the decision of the editors to accept such a layout of the work.
Response:
Actually, in accordance with the guidelines provided by the journal, we adhered to the prescribed Microsoft Word template outlined in the submission checklist to prepare our manuscript.
On a positive note, the very good documentation and illustrations of the obtained research results deserve to be emphasized.
Response:
We deeply appreciate your perspective.
